# War and Peace (WarAgent): LLM-based Multi-Agent Simulation of World Wars

## Abstract

This research explores the potential of Artificial Intelligence (AI) and Large Language Models (LLMs) in understanding and simulating complex human behaviors, specifically in the context of historical international conflicts. We introduce **WarAgent**, an LLM-powered multi-agent AI system, to simulate the decisions and consequences of participating countries in three specific historical conflicts. In addition, we propose standard evaluation protocols for LLM-based Multi-agent Systems (MAS) simulation. Our study provides a nuanced analysis of the strengths and limitations of current MAS systems in simulating complex collective human behaviors under diverse settings of international conflicts. The emergent interactions among agents in our simulations offer fresh perspectives on the triggers and conditions leading to war. Our findings offer data-driven and AI-augmented insights that can help redefine how we approach conflict resolution and peacekeeping strategies. While we acknowledge the potential of AI in providing data-driven insights, we caution against over-reliance and emphasize the need for careful interpretation in conflict resolution and peacekeeping strategies. The implications of this work extend beyond computer simulation, offering a potential avenue for using AI to better understand human history. Code and data are available at `https://anonymous.4open.science/r/WarAgent-0FF0`.

## 1 Introduction

In the wake of revolutionary advancements in Artificial Intelligence (AI), particularly the emergence of LLMs (OpenAI, 2023), we stand on the brink of a paradigm shift in computational social science research. In this study, we explore LLMs on social computing (Stauffer, 2002; Sun, 2006; Conitzer et al., 2020) and develop a novel framework of an LLM-based Multi-Agent System (MAS) (Nayak et al., 2023), specifically for the simulation of historical events. By creating a dynamic environment where agents of countries, each embodying the characteristics and decision-making processes of historical actors, engage in conflict or cooperation, we can explore the vast array of possibilities that could have shaped the evolution of international conflicts in the past, which have established the current global order. In this sense, our simulation provides multifaceted ways to explore these age-old puzzles that are central to the safety and warfare of humanity.

Traditional methods of studying conflict through historical analysis, while insightful, are limited by their static nature and the bias of hindsight. The application of simulations in social science has a rich history, but the fidelity and scope of these simulations have evolved dramatically. Early attempts were often limited by computational power and simplistic models (Smith, 1970; Hermann & Hermann, 1967). More recent simulations employ LLMs that can model complex behaviors and interactions, such as the virtual town simulation of human behavior (Park et al., 2023), the Werewolf game simulation (Xu et al., 2023), the auction arena simulation (Chen et al., 2023), and the complex task-solving simulation (Ge et al., 2023). These approaches have laid the groundwork for using AI to model more intricate systems, such as international relations and conflicts. However, there has not been research on how to apply these advanced technologies to simulate the nuanced and multifaceted nature of international diplomacy and war, which is where our research positions itself: **our paper aims to build the first LLM-based multi-agent system simulation of historical events**.

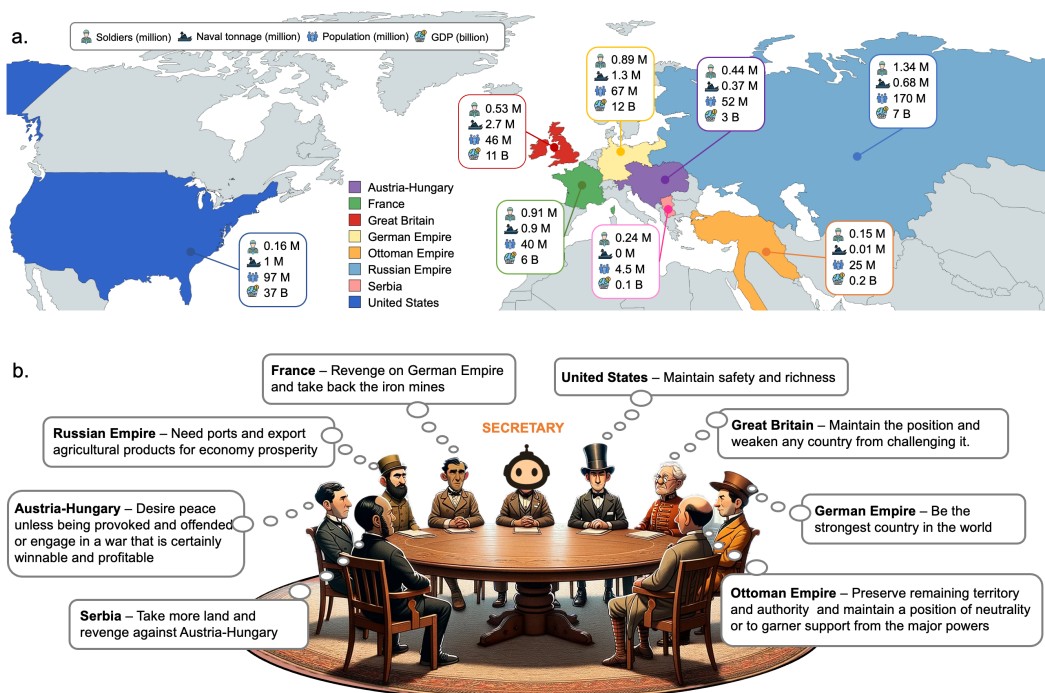

**Figure 1:** Demonstration of World War I Simulation Setting

At the core of our investigation are critical questions that challenge traditional understandings of historical conflicts. By addressing the following three questions, our research uses the microcosm of *World War I (WWI)*, *World War II (WWII)*, and *Warring States Period (WSP) in Ancient China* to gain insights about international conflict dynamics. The research questions include:

**RQ1, Simulation Effectiveness**  How effectively and accurately can LLM-based multi-agent simulations replicate the historical evolution of strategic planning and decision-making processes? By comparing the outcomes of our simulations with documented historical events and trends, we can measure the accuracy of the simulation. Only a system that upholds validity is capable of facilitating comprehensive analysis and addressing subsequent research questions. In addition, given extensive knowledge that LLM learn from pretrain data, we experiment on whether WarAgent is doing authentic simulation or merely retrieving information from its memory.

**RQ2, Casus Belli**  Are certain triggers of war more critical than others, and can they be identified through LLM-based multi-agent system simulations? We aim to isolate and analyze the various triggers of war to determine if certain triggers are more decisive than others in precipitating conflict.

**RQ3, War Inevitability**  Are historical inevitabilities truly unavoidable? We seek to uncover the conditions that lead to war (or peace) through LLM-based multi-agent system simulations. This question strikes at the heart of deterministic versus contingent views of history.

## 2 BACKGROUND AND RELATED WORK

**Multi-Agent Simulation**  Recent developments in Multi-Agent Systems (MAS) have opened new avenues in AI research. These systems, which coordinate and communicate among multiple agents, provide an innovative platform for examining emergent communication within agent communities tasked with specific problem-solving.

The existing MAS landscape is broadly categorized into three types: **reasoning-enhancement systems**, **NPC (Non-Player Character) multi-agent players**, and **production-enhancement systems**. In reasoning-enhancement systems, several noteworthy contributions have been made (Du

et al., 2023; Chan et al., 2023; Sun et al., 2023; Liang et al., 2023) where the concept of debate was proposed and multiple agents debate with each other in order to enhance the factuality, faithfulness, and reliability of the reasoning process. In the domain of NPC, several works such as generative agents (Park et al., 2023; Wang et al., 2023; Fu et al., 2023) simulate human behavior such as everyround behaviors and bargaining games for interactive applications. In production-enhancement systems, notable examples include MetaGPT (Hong et al., 2023), a specialized LLM application based on a multi-agent conversation framework (Li et al., 2023; Qian et al., 2023; Liu et al., 2023b) for automatic software development by assigning different roles to GPTs to develop software applications collaboratively.

*Our research extends the exploration of MAS to historical event simulations.* We leverage World War I (1914 - 1918), World War II (1939 - 1945), and the Warring States Period in ancient China (770 BC – 221 BC) as examples. Our research reveals how MAS can provide valuable insights into historical occurrences and hypothetical "what-if" scenarios, thus introducing a novel, quantitative dimension to the field of humanities. This work underscores the broader impact of AI beyond LLM itself and production, highlighting its potential to enhance our understanding of humanity. In responsible hands, AI can contribute significantly to larger social good and a brighter future for all.

**Traditional Historical Simulation Tools**   The academic study of history simulation has undergone several stages: **human simulation**, **human-program hybrid simulation**, and **computer simulation**. Dickson (Dickson, 2002) outlines a human simulation of the United States' journey toward participation in WWI in an educational scenario. In the 1960s, human-program hybrid systems were developed. The Inter-Nation Simulation model (Guetzkow et al., 1963) is employed in various studies (Hermann & Hermann, 1967), playing a pivotal role in simulating international conflicts. This method integrates human decision-making with computerized calculations, creating a dynamic hybrid simulation environment. In the 2000s, computing power was leveraged to build the next-generation historical simulation. The Army One Semi-Automated Forces OneSAF Objective System (OOS) (Tollefson et al., 2008) is a Computer-Generated Force (CGF) that has been designed to represent a full range of operations, systems and control processes from the entity level to the brigade level (Kelly et al., 2006). (Hill et al., 2004) presents a JAVA-based multi-agent modeling and simulation paradigm with game theory for an in silico historical analysis of the Bay of Biscay submarine war during WWII.

Our simulation leverages cutting-edge LLMs, which is presently recognized as the most promising AI paradigm. This marks our inaugural endeavor in employing MAS to model historical events, the first step in the field.

## 3   WARAGENT SETTING

This research centers on the simulation of international conflicts where each country is played by one LLM-based agent, specifically on the outbreak of WWI, WWII, and WSP. We focus on examining the dynamics of diplomacy and international relations and the likelihood of war initiation. Background details of the three historical events are presented in Appendix A. In this section, we introduce the basic settings in the system, the profile definitions of country agents, by elaborating on the dimensions of their profiles as well as the action space available to these agents.

**Country Agent's Profile**   The characterization of each agent in the model requires the delineation of a comprehensive profile. In the case of a country agent, this profile is to be constructed around six fundamental dimensions: Leadership, Military Capability, Resources, Historical Background, Key Policy, and Public Morale. Each aspect contributes to a multifaceted understanding of the agent's potential behavior and decision-making processes within the simulation. An example is presented in Appendix B.

**Action Space**   Our framework is designed with the specific objective of examining the onset of wars, and accordingly, it includes a suite of actions that shape international relations between countries, categorized into seven distinct groups: **Wait for action**–An agent may opt to take a passive stance during certain rounds, observing the actions of others and changes in the broader context. **General mobilization**–This action involves preparing a nation's military forces for potential conflict, a precursor step required before engaging in war. **Declare war**–A country can formally initiate

hostilities against another. **Military alliance**–This is a formal agreement between two or more nations to provide mutual support in case of conflict, bringing with it a shared responsibility for collective defense and strategic cooperation. An agent can request an alliance for mutual defense, which others can accept or reject. **Non-intervention treaty**–This is a diplomatic agreement where signatory states commit to abstain from interfering in each other's internal affairs, entailing a responsibility to respect sovereign integrity and political independence. The procedure of signing a non-intervention treaty is similar to forming a military alliance: an agent first requests; the target agent may accept or reject. **Peace agreement**–This is a negotiated settlement between conflicting parties that formally ends hostilities and establishes the framework for future relations. Parties in conflict can propose and either accept or reject a peace agreement to conclude hostilities and outline the basis for their future relationship, with options for publication or betrayal of the agreement. **Send message**–Aside from formal actions, agents can communicate informally through messages to discuss various matters.

Each action is defined to be public or non-public. Public actions, such as "Publish Military Alliance" are broadcasted to all country agents within the simulation, reflecting actions in real-world politics that are openly disclosed and known internationally. On the other hand, private actions such as "Request Military Alliance" are communicated only to the targeted country, mirroring confidential or behind-the-scene diplomatic maneuvers in real-world international relations. By implementing this feature, the simulation ensures that each country agent operates based on its own knowledge, which may be complete or limited depending on the nature of the actions taken by others.

## 3.1 ANONYMIZATION OF HISTORICAL EVENT

Since LLMs are pre-trained on vast corpora of textual data with a substantial repository of knowledge, there exists a possibility that these models recall and subsequently reproduce the actual historical trajectory during simulation. To circumvent this potential issue, we anonymize country names and introduce minor modifications to historical facts. These modifications are designed to ensure that these alterations do not materially affect the simulation's efficacy, thereby maintaining the integrity and originality of the simulation outcomes (Details in Appendix C). We compare anonymized simulations and de-anonymized simulations by experiments and verify the effectiveness of the method and also the authenticity of our simulation.

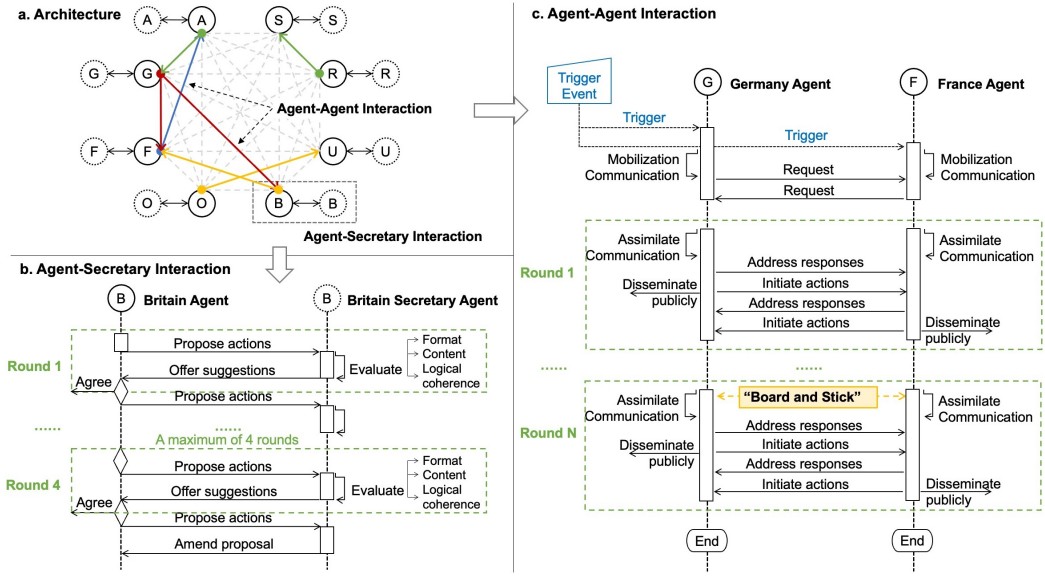

**Figure 2:** WarAgent Architecture (a) WarAgent architecture. (b) Agent-Secretary interaction. (c) Agent-Agent interaction.

## 4 WARAGENT ARCHITECTURE

This section provides a comprehensive introduction to the architecture of the WarAgent, detailing its core components and the information flow among agents. WarAgent is built upon four foundational building blocks: (1) Country agents, (2) Secretary agents, (3) Board, (4) Stick. The section then shifts to explore the mechanisms of information exchange within the MAS, particularly focusing on (1) Agent-Secretary interaction and (2) Agent-Agent interaction. This exploration aims to study how agents communicate, interact, and share information.

### 4.1 BUILDING BLOCKS

**Country Agents** Each country agent is defined by its corresponding country profile. In each round, the agent reacts to the current situation by generating actions available from the action space, guided by meticulously structured prompts which directs the agents to analyse who their potential enemies and potential allies are based on the principle of "the enemy of an enemy is a friend" in order to maximize its self-interest.

**Secretary Agents** While LLMs are powerful tools in facilitating MAS, they are not infallible. They often exhibit limitations, such as a tendency towards hallucination and a lack of solid logical reasoning, particularly in long contexts with complex, extended scenarios (Liu et al., 2023a; Yang & Ettinger, 2023). Therefore, the presence of a secretary agent serves as a necessary safeguard, providing a fundamental check against hallucinations. Each country agent has a designated "secretary agent" to verify the appropriateness and basic logical consistency of their actions. This role is twofold. Firstly, the secretary agent ensures that all action generated are valid in the provided action space, including the correct name of the action and correct formatting of inputs. Secondly, the agent is responsible for verifying the basic logical coherence of these actions. For instance, it would be illogical and inadmissible for Austria-Hungary to "Accept a Military Alliance" from Britain if Britain had not initiated the process by sending a "Request for Military Alliance" to Austria-Hungary.

**Board** The Board is designed to manage international relationships. It acts as a dynamic recording platform that collects and displays the relational dynamics of the ongoing situation in each simulation round. It further ensures that the agents' decisions are based on the most up-to-date available information. As shown in Figure 2(a), the Board class can track the following four types of international relationships showing by different colors: **War Declarations**: indicate conflicts or wars between countries, marked in red. **Military Alliances**: denote formal military partnerships between countries, marked in green. **Non-intervention Treaties**: represent agreements of non-interference in international affairs, marked in blue. **Peace Agreements**: represent formal agreements to cease hostilities and maintain peace between countries, marked in yellow. In this example: German Empire declares war on Great Britain and France; Serbia and Russia are Military Alliance; Austria-Hungary and France sign a Non-Intervention Treaty; the United States and Ottoman Empire reach a Peace Agreement.

**Stick** The Stick functions as an internal record-keeping system for each country that represents the domestic status such as whether a country's army is mobilized. It can help to ensure that the country agent's actions align with its predefined protocols and standards.

### 4.2 AGENT INTERACTION

As shown in Figure 2(a), the system's agent interaction is bifurcated into two primary segments: (1) internally, each country agent interacts with its corresponding secretary agent in every round, and (2) externally, each country agent interacts with other country agents across multiple rounds.

**Agent-Secretary Interaction** Figure 2(b) shows the Agent-Secretary interaction. In each round of the simulation, there is a designated interaction between each country agent and its corresponding secretary agent. The country agent presents a proposed plan of action, which the secretary agent then evaluates for format, content, and logical coherence. Should the secretary agent find discrepancies or areas for improvement, it offers suggestions and engages in a dialogue with the country agent for revision. This iterative process is capped at a maximum of four rounds of exchanges. If agreement is not reached within these exchanges, the secretary agent takes the initiative to directly amend the proposal. This dynamic of "country agent–secretary agent" internal interaction is a consistent feature across all external rounds and is applicable to all country agents.

**Agent-Agent Interaction**  Figure 2(c) illustrates the Country Agent-Country Agent interaction. It should be noted that the secretary agents do not participate in the interactions that occur between country agents. For clarification, "agent" in this context specifically denotes a country agent.

In our framework, the very initial actions proposed by each agent are precipitated by a *triggering event* (denoted in the blue box in Figure 2(c)). Within the historical context, a triggering event refers to an incident that initiates a sequence of reactions among various countries, ultimately culminating in a significant historical development. In our simulation, a triggering event serves as the initial scenario to which all agents respond. For instance, in the case of WWI, the assassination of Archduke Franz Ferdinand of Austria-Hungary is widely recognized as the triggering event (Mombauer, 2013).

In the first round of the simulation, each participating agent reacts to the trigger event in unison by variety of actions such as general mobilization and interactive communication through assorted messages and requests directed at all other agents. Subsequently, agents assimilate communications from the preceding phases, specifically those aimed directly at them or those disseminated publicly. We leverage the Board to record relationships such as "Military Alliance" formed between agents and Stick to record agent-internal information such as "General Mobilization". Note that during each round, a multitude of new communications and evolving scenarios emerge as all country agents engage in simultaneous interaction, generating a substantial volume of text so that the agents can potentially become overwhelmed and lost in the middle. To address this challenge, agents rely on the Board and Stick for latest situation. Thus for each round, agents only have access to the one-previous round of conversation and information on the Board and Stick, turning the multi-turn conversation into a semi-Markovian process.

## 5 EXPERIMENT DESIGN AND RESULTS

Our experiments start with verifying the effectiveness of WarAgent in simulating complex historical events. We utilize three strong models as backbone models for the experiments: GPT-3.5-turbo-1106, GPT-4-1106-preview, and Claude-2. All experiments are conducted using all models unless specified otherwise.

### 5.1 EVALUATION METRIC

Here are the evaluation methods for each research question: for RQ1, we mainly use board-based accuracy to compare the simulated results with historically true results, and use human analysis to manually verify the logical consistency across rounds in the simulation. For RQ2 and RQ3, we apply counterfactual analysis by altering the original settings and conducting observational analysis on the difference in the simulation results.

Here we introduce more details for the three aspects of Board-based Accuracy: Given the often contentious and debatable nature of historical events, this evaluation adopts descriptions and summaries from a widely recognized historical masterpieces (Keegan, 2014; Churchill, 1948; Chan, 1999) as basis for historical facts. This approach provides a standardized and widely accepted reference point for comparing simulated situations with real historical events. For *Alliance Accuracy Score:* Alliance formation is a transitive process, which means that if country A forms an alliance with country B and B forms an alliance with country C, then countries A and C naturally become alliances[1]. Thus alliance formation among a group of countries can be seen as a partition of a set. To evaluate the congruence of simulated alliances with historical alliances, we employ the **mutual information score of two partitions** (Vinh, 2010). This metric offers a quantifiable measure of the similarity between the two sets of partitions—the simulated and the historical. We adopt the standard implementation from SciPy in Python. For *War Declaration* and *General Mobilization Accuracy Score:* We employ the Jaccard set similarity index as the primary metric for calculating the accuracy of the simulation in these two domains. The **Jaccard index** (Rajaraman & Ullman, 2011), a widely recognized statistical tool for measuring the similarity and diversity of sample sets, quantifies the degree of commonality between the respective sets. This method involves comparing the similarity between two sets of pairs, specifically in the context of war declarations, and two sets of singletons in the case of general mobilization.

---

[1]https://www.defense.gov/News/Feature-Stories/story/Article/1684641/alliances-vs-partnerships/

## 5.2 RQ1 RESULT

In this subsection, we present the accuracy results for the three historical scenarios. We then discuss based on two experiments the authenticity of simulations , *i.e.* the simulations are generated based on LLM's reasoning ability rather than memory. Human evaluation and further error analysis are presented in Appendix D.1.1 and D.1.3.

For **WWI**, our simulation focuses on the time frame from June 28th to August 4th, 1914, for evaluating the accuracy of the simulated alliances and war declarations. The selection of this period is grounded in historical significance. Notably, the Battle of Liège on August 6th, 1914, is recognized historically as the inaugural major battle of World War I. For **WWII**, our simulation focuses on the time frame until September 18th, 1939, when Britain and France had declared war on Germany, and Soviet Union (Russia)'s armies reached Vilnius and Brest-Litovsk, indicating the major powers' involvement in WWII. **WSP** is a historical phase characterized by the ever-evolving alliance relationships among the seven states and the relentless declarations of war. To accurately assess the military alliances during this tumultuous period, we choose the event in 323 BC in response to the aggressive expansionism of the Qin state against Wei State, which serves as the ground-truth for our analysis[2]. The ground-truth based on the above historical facts are presented in Appendix D.1.2. We conduct 7 separate simulation runs and report the average and standard deviation of accuracy.

| Model | Scenario | Evaluation Aspects | | |
|---|---|---|---|---|
| | | alliance | war | mobilization |
| GPT-4 | WWI | 77.78 (±5.19) | 54.60 (±3.82) | 92.09 (±4.89) |
| | WWII | 73.69 (±14.80) | 45.89 (±4.83) | 75.48 (±11.34) |
| | WSP | 78.32 (±3.02) | - | 100.00 (±0.00) |
| Claude-2 | WWI | 60.69 (±15.14) | 16.76 (±3.31) | 97.96 (±5.00) |
| | WWII | 42.86 (±18.82) | 42.06 (±11.09) | 75.25 (±7.18) |
| | WSP | 51.09 (±15.63) | - | 67.00 (±14.30) |
| GPT-3.5 | WWI | 43.28 (±26.28) | 7.83 (±8.11) | 78.06 (±14.69) |
| | WWII | 12.18 (±13.34) | 0.00 (±0.00) | 65.25 (±10.95) |
| | WSP | 68.29 (±18.10) | - | 78.34 (±16.70) |

**Figure 3:** Board-based mean and standard deviation

Table 3 presents the evaluation results: simulation reaches high accuracy with low variance on alliance and mobilization for GPT-4 simulations while the results of GPT-3.5 and Claude-2 are worse due to their weaker reasoning ability. We observe that GPT-3.5 and Claude-2 lack comprehension of the principle that "the enemy of an enemy is a friend", thereby failing to identify appropriate countries for forming alliances or declaring war. Conversely, the accuracy of our simulated war declarations, including those generated by GPT-4, remains relatively low. This is largely due to the uncertainty in determining against which country within the same enemy military alliance a war declaration should be made. For example, the Britain agent may declare war against Austria-Hungary or German Empire in WWI simulations, whereas historically, Britain only declared war against German Empire. But in general, in all our simulations, the world war breaks out without exception for all scenarios, which is consistent with historical facts. Detailed analysis is presented in Appendix D.1.3.

To demonstrate that the **simulation is authentic** which does not merely copy from memory but do analysis and reasoning, we conduct two experiments: (1) Injecting counterfactual knowledge into the model and see whether the model will be misled: if the simulation copies from memory, the simulation results should be affected by the injected knowledge (Zhu & Li, 2023). In experiment, we inject counterfactual history where Austria-Hungary and Serbia achieved peace after the assassination of Archduke Ferdinand in 1914 by finetuning on these counterfactual conversations for 3 epochs. Examples of these counterfactual conversations are shown in Appendix D.1.4. (2) Simulation using non-anonymized setting aiming to replicate history. The premise is that if our anonymized simulations were to draw directly from historical events, they should exhibit characteristics similar to those observed in the non-anonymized simulations. Otherwise, they should exhibit certain discrepancies from history-replication simulations.

**Counterfactual Injection** The experiment is conducted on GPT-3.5 and results are presented in Table 1. We add the metric "global war" which is evaluated to be True if major powers (Britain, France, German Empire, Russia, Austria-Hungary) are all involved in war. We conduct 7 simulations using the counterfactual-tuned model and observed that it eventually still leads to the global

---

[2]As war declaration were non-stopping and constantly evolving during WSP, we do not evaluate war-declaration accuracy for WSP

war scenario, same as non-counterfactual-tuned model. This experiment shows that the explicit integration of a peaceful counterfactual history of 1914 does not affect the simulated result of a global outbreak of war, indicating that the simulation does not rely on the dataset that is used to train, *i.e.* its memory.

| Model | Evaluation Aspects | | | |
|---|---|---|---|---|
| | alliance | war | mobilization | global war |
| before counterfactual | 43.28 (±26.28) | 7.83 (±8.11) | 78.06 (±14.69) | 100.00 (±0.00) |
| after counterfactual | 31.02 (±20.30) | 15.18 (±6.28) | 62.12 (±19.48) | 100.00 (±0.00) |

**Table 1:** Accuracy of simulation before and after injecting counterfactual knowledge to GPT-3.5

**Deanonymization** We deanonymize the content and instruct country agents to rely on their memory and replicate on historical records at the breaking-out time of WWI. All simulations are run for 7 times. The results are presented in Table 4.

| Model | Evaluation Aspects | | |
|---|---|---|---|
| | alliance | war declaration | mobilization |
| GPT-4 | 97.43 (±0.00) | 14.17 (±5.13) | 100.00 (±0.00) |
| Claude-2 | 97.43 (±0.00) | 18.72 (±9.82) | 100.00 (±1.12) |
| GPT-3.5 | 97.43 (±0.00) | 13.19 (±7.33) | 100.00 (±0.00) |

**Figure 4:** Accuracy of deanonymized WWI simulation

The results show that all of the three models perform very accurately and consistently on military alliance and general mobilization. The simulation results also converge at a very early stage of the simulation–the 2nd round. However, it performs poorly on war declaration. We inspect the results and notice that this is because–as these agents completely rely on historical facts–the simulation presents almost all war declaration relationships during the whole time period of WWI instead of at the breaking out time of the war. Thus, its result is evaluated to be even worse than that in our simulation. In addition, we notice that de-anonymized simulation never includes formation of Non-intervention Treaty between any country, which is historically correct but inconsistent with our anonymized simulation where Non-intervention Treaty is achieved very often between two countries that have no intention to involve into conflicts against each other. Visualization of simulation dynamics can be found in Appendix D.1.5. Table 5 summarizes the differences between anonymized simulation (denoted as A) and de-anonymized simulation (denoted as De-A), implying the authenticity of our simulation.

| Features | A | De-A |
|---|---|---|
| Close alignment with history | | ✓ |
| Fast convergence to historical scenario | | ✓ |
| Consistent simulation across models | | ✓ |
| Construct historically-non-occurring relations | ✓ | |
| Stable simulation across random seeds | | ✓ |

**Figure 5:** Summary of difference between anonymized simulation and de-anonymized simulation

**Agent Network Dynamics in Simulation** To figuratively demonstrate the diffrerence between anonymized simulation that we adopt and deanonymized simulation, visual representation for the dynamics between countries in WWI generated by GPT-4. Figure 6 depicts a six-round evolution in a round of simulation, characterized by changing relationships between different country agents (B, F, G, A, R, S, U, and O). Each cell within the board represents the relationship between two entities, with the rows and columns designated by the same set of labels indicating bilateral interactions. We show the true relationships without any third-part agent guessing. *Round 1: Initial State* On round 1, the board is primarily in a default state (yellow), indicating a neutral or baseline state of affairs between all entities. *Round 2: Formation of Alliances and Non-intervention Pacts* By round 2, we see the emergence of military alliances (blue, "&") and an increase in non-intervention treaties. *Round 3: Escalation and War Declarations* The third round is marked by the first instances of war declarations (red, "×"). These are concentrated among specific entities rather than widespread, indicating targeted conflicts rather than a general state of

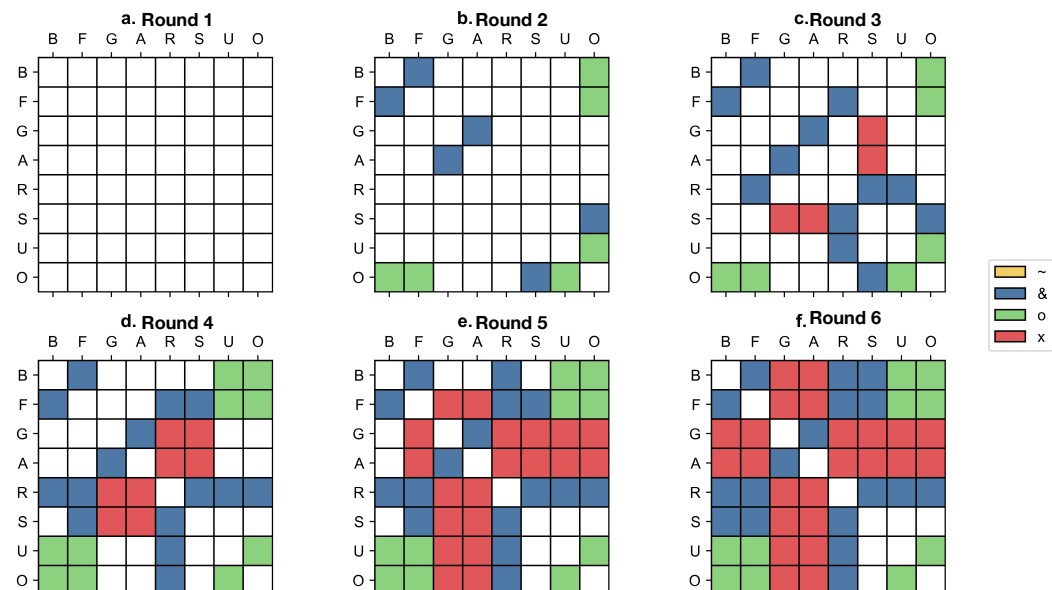

**Figure 6:** Examples of a Six-round Evolution of Board. The notations stands for Default (white); Peace Agreement (yellow), "˜"; War Declarations (red), "×"; Military Alliances (blue), "&"; Non-intervention Treaties (green), "o".

war. *Round 4 - 5: Intensification of Conflicts* On Round 4 and 5, the number of war declarations has increased, showing an intensification of conflicts. The spread of red cells indicates that the simulated world is moving towards a more conflict-prone phase. *Round 6: Peak of Conflict* Round 6 shows the peak of conflict, with war declarations becoming the predominant state across the board. This could reflect a world war scenario where conflicts have spread and the majority of entities are engaged in warfare.

**Network Dynamics for De-anonymized Simulation** We also present the de-anonymized simulation result generated by GPT-4 in Figure.7, showing how de-anonymized simulation can quickly align with historical facts. *Round 1: Initial State* On Round 1, the board is primarily in a default state (yellow), except that Austria-Hungary and Serbia are already at war. *Round 2: Formation of Alliances* By Round 2, we see the emergence of military alliances (blue, "&") between Britain, France, Serbia, and Russia (the Allied Powers,), as well as German Empire, Austria-Hungary, and Ottoman Empire (the Central Powers). We also see more wars between countries belonging to the Allied Powers and the Central Powers. *Round 3-6: Peak of Conflict* The remaining rounds witness escalation of wars, where all countries are involved into wars and the scenario stays unchanged during Round 4, 5, and 6. Notice that in the six rounds of simulation, no non-intervention treaty is signed between any two countries, as in the real history.

**Summary** This six-round evolution of the board suggests a fast alignment of simulation and real history, where country agents shift from a state of neutrality to forming alliances and starting conflict in basically two rounds. This contrasts with the network dynamics in Figure.6, demonstrating the difference between anonymized simulation and de-anonymized simulation.

## 5.3 COUNTERFACTUAL ANALYSIS: RQ2 & RQ3

We conduct counterfactual analysis mainly on WWI. Below are the main experimental results, including Casus Belli and War Inevitability. Main findings are summarized here while detailed analysis are presented in Appendix D.2 and D.3.

**Casus Belli** involves examining various alternative trigger events to determine their impact on global conflicts. We conduct experiments on WWI scenario by constructing three non-factual trigger events with varying intensity and simulate the results using GPT-4 (Appendix D.2).

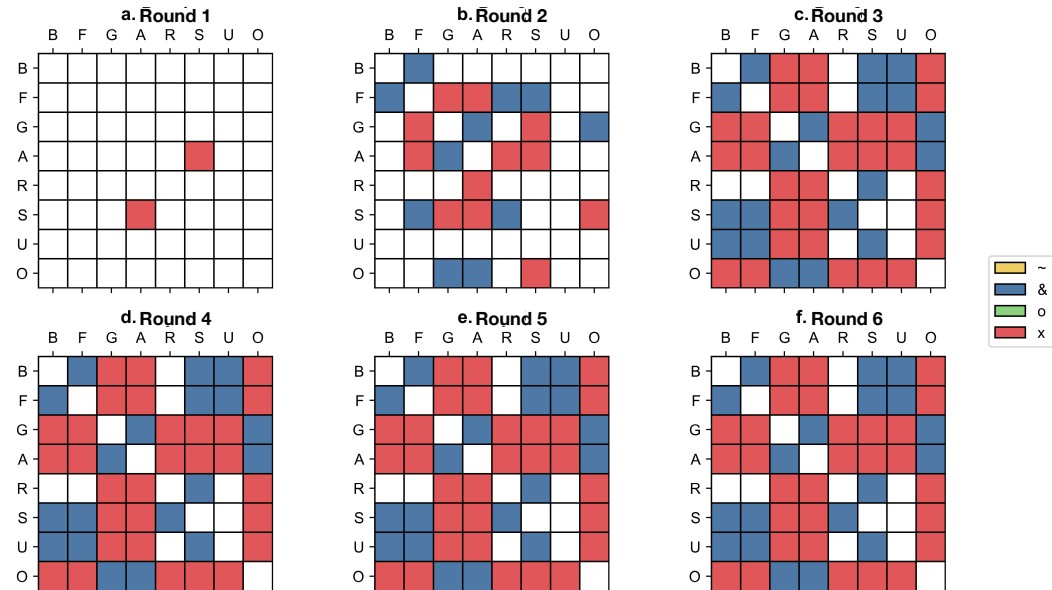

**Figure 7:** Examples of a Six-round Evolution of Board. The notations stands for Default (white); Peace Agreement (yellow), "~"; War Declarations (red), "×"; Military Alliances (blue), "&"; Non-intervention Treaties (green), "o".

The first trigger is Null trigger, characterized by the absence of conflict. The simulations generate situation where all involved countries are ready for war, yet no actual "hot" war ensue. This outcome indicates that the major powers maintained a balance of power, remaining on the brink of war but not crossing into open conflict, thereby creating a scenario akin to a cold war. The second trigger, termed the "Anglo-German Naval Incident," represents a medium level of conflict intensity, involving significant but not critical diplomatic or military engagements: in 1 out of 3 simulations global war occurs. The third and most intense trigger is the "Austria-Russia Conflict over the Dardanelles Strait," which depicts a high-conflict scenario directly engaging two major European powers: global war occurs in 1 simulation and regional conflict occurs in another. This gradation in conflict intensity provides a spectrum for assessing the impact of varying degrees of geopolitical tensions on WWI's potential outbreak of (Schroeder & Schroeder, 2004; Tenembaum, 2015).

**War Inevitability** study includes how different values of country profiles aspects can impact the likelihood of war. Experiments are conduct on WWI using GPT-4 (Appendix D.3). Experiments reveal that historical context, key policies, and public morale significantly influence a nation's inclination towards war. For example, the Franco-Prussian War's legacy, which involved territorial losses for France and the unification of the German Empire, fueled enmity and a desire for retribution in France, setting the stage for future conflicts. While military capability and resources are important, they do not solely dictate war engagement. Historical and diplomatic contexts, such as alliances and defense pacts, were found to be more decisive in the actions of nations such as the German Empire and France, even when military strength varied. This highlights the complex interplay of various factors in shaping a nation's military decisions.

## 6 CONCLUSIONS

The WarAgent simulation system is the first attempt to utilize LLM for understanding and evaluating the dynamics of international conflicts, showcasing the LLM-based multi-agent AI systems' ability of prototyping and analyzing complex human behaviors. Comparing various Casus Belli settings, our experiments reveal that even minimal or "null" triggers can spiral into situations reminiscent of the Cold War, highlighting the often-inevitable progression towards war. This is further supported by the War Inevitability experiments, through counterfactual alterations in national settings, suggesting that deviations in national policies are necessary to divert from the path to conflict. Findings in paper open up a realm of possibilities for preventing conflict, highlighting the role of strategic policy intervention in changing what might seem like preordained paths to war.

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

LIMITATIONS

This paper introduces groundbreaking research in the use of LLM-based multi-agent AI systems. However, there are several limitations:

## 6.1 DESIGN OF MECHANISM

The present study conducts simulations encompassing brief durations from the precipitating event to the onset of warfare. These periods include approximately one and a half months for World War I (June 28th to August 4th, 1914), half a month for World War II (September 1st to September 18th, 1939), and a maximum of one year for the Warring States Period (WSP) around 323 BC, given the paucity of precise date documentation. Consequently, the interaction mechanism elucidated in this paper pertains solely to these specific, brief timeframes, rather than offering a comprehensive simulation of historical events. For the execution of alternative historical simulations, the implementation of more intricate designs would be requisite.

## 6.2 CAN WE TRUST SIMULATION RESULTS?

Since the early 20th century, computers have played a pivotal role in social science research, particularly in social simulation. In these simulations, computers emulate human reasoning processes and the resultant scenarios. The objective is to generate simulations that allow for inductive analysis based on either a strictly defined set of rules or, as in our case, the intelligence derived from LLMs trained on extensive human-generated documents and corpus.

However, criticisms have been around for computer simulation and we list 4 points beloe:
**Simplicity**: The view that simulations are overly simplistic representations of human society.
**Limited Insight**: Concerns that simulations fail to enlighten researchers about unprogrammed human interactions.
**Relevance to Real World**: The difficulty in relating abstract simulation results to the complexities and variations of actual societies.
**Verification Challenges**: The notion that simulation results are unverifiable and thus meaningless.

In response to these criticisms, our stance is as follows:

**Comparison with Social Science Theories**: Contrary to the criticism that simulations oversimplify social phenomena, we argue that traditional social science theories often present an even more simplified view of social dynamics. These theories are frequently derived from linear models or small-scale laboratory experiments that fail to capture the intricate dynamics produced by interactions among large populations. In contrast, our AI and LLM-based simulation results offer a more complex and nuanced understanding of these dynamics, providing a richer and more comprehensive model of social behavior.
**Value in Unverifiability**: The challenge of verifying simulation results with real-world experiments or empirical data does not diminish their value. Given the inherent difficulties in conducting large-scale social experiments, simulations emerge as a crucial tool for exploring hypothetical scenarios. They enable researchers to model and analyze the potential outcomes of various policies or social changes, offering insights that would be otherwise unattainable due to ethical or practical constraints.
**Role in Policy Discussion**: The difficulty in verifying social science theories does not render them useless. Discussions about societal policies, such as crime approaches which rely on unverified theories, are essential for democratic governance and policy formulation. These debates often rely on theoretical frameworks and hypotheses without possibility of being verified quantitatively in real world, still hold great values to the society.
**Simulations as Suggestive Tools**: We posit that simulation results should be interpreted as informative suggestions or rationales rather than definitive conclusions. These results provide policymakers, historians, and students with valuable hypothetical insights into the potential outcomes of various actions and policies. By offering a range of possible scenarios and outcomes, simulations serve as a useful tool in decision-making processes, aiding in the evaluation of different strategies and policies. It is ultimately up to human judgment to interpret these results and make informed decisions. Simulations, therefore, should be viewed as valuable aids in the decision-making process, contributing to a deeper understanding of complex social dynamics.

In summary, while acknowledging the limitations and criticisms of computational social simulations, we emphasize their significant role as a complementary tool in computational social science research and policy analysis, providing unique insights and perspectives that enhance our understanding of complex social systems.

# A HISTORY BACKGROUND OF WWI, WWII, WARRING STATES

In this section, we commence by introducing three significant historical events that form the background of our MAS simulation study.

**World War One (WWI)** was a global conflict that lasted from 1914 to 1918. It was primarily fought in Europe but involved countries from around the world. The war started following the assassination of Archduke Franz Ferdinand of Austria-Hungary, which led to a series of political and military alliances being activated.

The major powers involved were divided into two main alliances: the Allies (originally composed of France, Russia, and the United Kingdom, later joined by Italy, Japan, and the United States) and the Central Powers (mainly the German Empire, Austria-Hungary, the Ottoman Empire, and Bulgaria). The war was characterized by trench warfare on the Western Front and fluid movements of armies over large areas on the Eastern Front, seeing significant use of new military technologies like machine guns, tanks, and chemical warfare. The Treaty of Versailles, signed in 1919, officially ended the war but imposed heavy reparations and territorial losses on the German Empire.

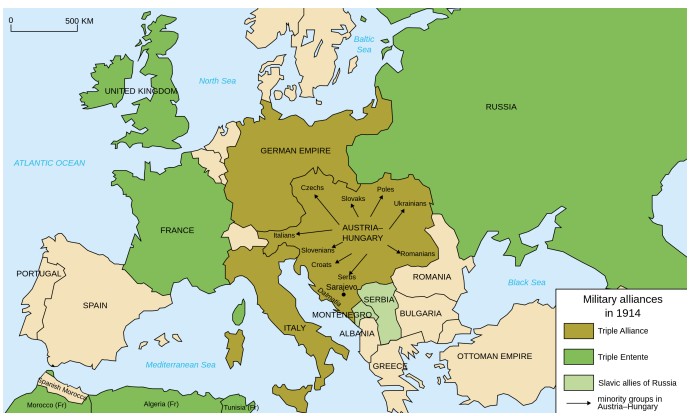

**Figure 8:** Map of World War I

image from `https://en.wikipedia.org/wiki/World_War_I`

**World War Two (WWII)** was a global conflict that lasted from 1939 to 1945, whose origins were linked to unresolved issues from WWII and the rise of fascist regimes in the German Empire, Italy, and Japan. It was the most widespread war in history and directly involved more than 100 million people from over 30 countries. German Empire's invasion of Poland in September 1939 prompted Britain and France to declare war on the German Empire, marking the beginning of WWII.

The major participants were divided into two opposing military alliances: the Allies and the Axis. The Allies primarily included the United Kingdom, the Soviet Union, the United States, and China. The Axis was led by the German Empire, Italy, and Japan. WWII saw the first and only use of nuclear weapons in war, with the United States dropping atomic bombs on the Japanese cities of Hiroshima and Nagasaki in August 1945. The war in Europe ended with the unconditional surrender of the German Empire in May 1945, but it continued in the Pacific until Japan's surrender in August 1945 following the atomic bombings.

**Warring States Period (WSP) in Ancient China** was a time of intense warfare and political turmoil in ancient China that lasted from 475 BCE to 221 BCE, marking the final centuries of the Zhou Dynasty. This era followed the Spring and Autumn period and led up to the unification of China under the Qin Dynasty.

The Zhou king's authority diminished during this period, and regional warlords or states became increasingly powerful and independent. These states were constantly at war with each other, vying for dominance. The period is characterized by the existence of seven major states: Qin, Chu, Yan, Han, Zhao, Wei, and Qi. Each state had its own ruler and army, and alliances between states were frequently made and broken. The WSP saw significant military advancements. Iron weaponry became more common, cavalry units were introduced, and large infantry armies were mobilized. The period ended with the state of Qin, under Qin Shi Huang, defeating all other rival states and

unifying China in 221 BCE. Qin Shi Huang became the first emperor of a united China, marking the Imperial Era's start and the feudal system's end.

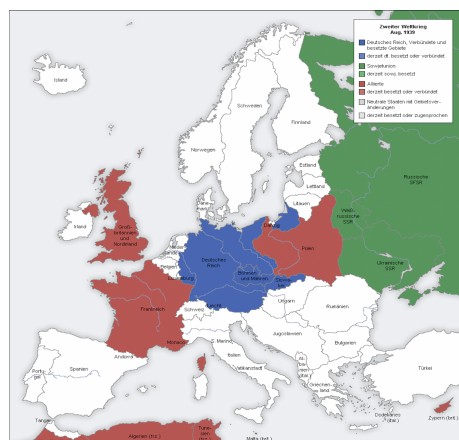

**Figure 9:** Map of World War II

image from https://en.wikipedia.org/wiki/World_War_II

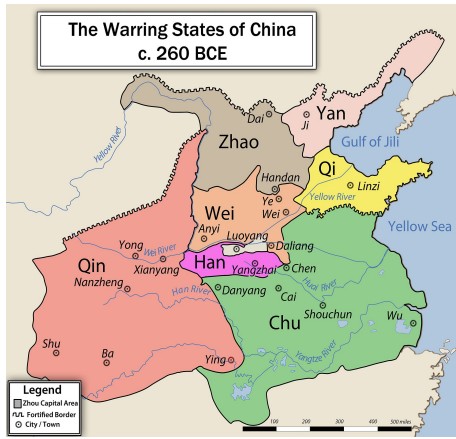

**Figure 10:** Map of Warring States Period

image from https://en.wikipedia.org/wiki/Warring_States_period

## B  PROFILE DEFINITION WITH EXAMPLE

**Leadership** encompasses the political institutions responsible for decision-making within a nation, contextualized by specific historical periods. For instance, before WWI, Britain exemplified a constitutional monarchy replete with democratic structures, distinguished by pragmatic and stoic leadership. In contrast, Prussia operated under an autocratic imperial regime, with a foreign policy geared toward aggression and military expansion, pivotal to its national ethos.

**Military capability** comprises quantitative data such as the size of its standing army, naval tonnage, and a qualitative assessment of its overall military strength, including any particular dominance in specific branches, such as naval or aerial forces. A conclusion regarding military might is essential, as it correlates with a nation's propensity to engage in or declare war; countries with robust military capabilities are typically less hesitant to partake in military conflicts.

**Resources** encompass critical elements such as geography, population, Gross Domestic Product (GDP), terrain, and climate conditions. Population size and GDP are particularly salient indicators of a country's strength and serve as pivotal considerations in the strategic decision-making processes of the agent. These factors provide a measure of the nation's potential economic and logistical support for its objectives, influencing its capacity to project power and sustain military and political efforts.

**Historical background** incorporates the legacy of prior conflicts of interest and unresolved issues between nations, which can considerably influence current policies. Historical enmities and territorial disputes often leave indelible marks on a nation's current posture and potential alignment within the global arena. An illustrative case is the aftermath of the Franco-Prussian War, where France's loss of the Alsace-Lorraine region—an area rich with iron mines critical to its industrial development—engendered a fervent desire for retribution against Prussia. They shape a country's strategic alliances and influence its broader diplomatic and military engagements.

**Key policy** outlines the principal objectives pursued by nations. For instance, in the historical context, post-unification Prussia harbored ambitions of becoming the preeminent European power, seeking to eclipse Britain's supremacy in territories and colonies. Conversely, Britain was intent on maintaining its status as "the empire on which the sun never sets." This led to strategic policies such as the "Arms Race Act," a stipulation mandating that for every warship built by Prussia, Britain would respond by constructing two, exemplifying the competitive dynamics and the centrality of naval power to their geopolitical strategies.

**Public morale** reflects the populace's sentiment, which can directly or indirectly influence a country's action. For instance, a surge in nationalism within Serbia, despite its smaller size and limited military strength, fostered a bellicose attitude among its people. In contrast, despite its wealth and industrial prowess, the United States exhibited a stoic and isolationist disposition, with a general aversion to entering wars. While the impact of public morale on a nation's decisions can be moderated by the type of leadership in power, it undeniably plays a role in shaping the nation's policies and actions.

An example profile definition for Britain before WWI is presented here:

```
1  ## Britain profile
2
3  # Leadership for Britain
4  (1) A constitutional monarchy with significant democratic institutions,
       characterized by the pragmatic and stoic governance
5
6  # Military Capability for Britain
7  (1) Standing army population: 0.53 million soldiers
8  (2) Naval tonnage: 2.7 million, the strongest naval force in the world,
       whose tonnage is more than the sum of the second and third strongest
       naval force tonnage in the world
9
10 # Resources for Britain
11 (1) Geography: Small island to the west of France, German Empire, Austria
       -Hungary, Russia with large colony
12 (2) Population: 46 million
13 (3) GDP: 11 billion, consisting 13.6% of the whole world
14 (4) Terrian: Characterized by rolling hills, green fields, and rugged
       coastlines, often dampened by its maritime climate
15 (5) Weather: temperate maritime weather, often cloudy, rainy, and cool
16
17 # History Background for Britain
18 (1) Currently, Britain is the strongest country with most colony in the
       world
19
20 # Key Policy
21 (1) As the strongest country, Britain aims at maintaining the position
       and weakening any Francerom challenging it, such as German Empire
22 (2) For every warship being constructed by German Empire, Britain will
       construct two warships
23
24 # Public Morale for Britain
25 (1) public morale is high with a sense of patriotic duty and confidence
       in a quick victory
```

**Listing 1:** A demonstrative profile of Britain

## C  SETTING ANONYMITY

Country name anonymity:

```
1  Britain --> Country B
2  France --> Country F
3  German Empire --> Country G
4  Austria-Hungary --> Country A
5  Serbia --> Country S
6  Russia --> Country R
7  US --> United States
8  Ottoman Empire --> Country O
```

Location anonymity:

```
1  Alsace-Lorraine --> two iron mines
2  Dardanelles Strait --> Allison Strait
```

Event anonymity:

```
1  Assassination of Archduke Franz Ferdinand of Austria-Hungary
2  -->
3  Assassination of King of Country A
```

# D RESULTS

## D.1 SIMULATION EFFECTIVENESS

We conduct 7 simulation runs and present the human evaluation results of these runs. Considering the space limitation, we only present some overall results during the simulation and analyze the rationale of some special observations, leaving one run of the complete actions lists detailed in Appendix E.

### D.1.1 HUMAN EVALUATION

We present the observations of the simulation results using WWI simulation results generated by GPT-4 as examples in the following three aspects (i.e., Military Alliance, War Declaration, and Non-Intervention Treaty[3]):

- **Military Alliance**: For 100% of the simulation results, we observe consistent alliances formed between Britain and France, between German Empire and Austria-Hungary, and between Serbia and Russia. From the perspectives of Britain and France, the German Empire was viewed as a potential adversary, primarily because of its assertive expansionist policies. German Empire's strategic decision to forge an alliance with Austria-Hungary was influenced by a confluence of linguistic and ethnic commonalities, as well as a range of strategic and political considerations. Key among these was the mutual objective of diplomatically isolating France and establishing a united front to counterbalance the potential threat posed by Russia. Concurrently, the alliance between Serbia and Russia appeared to be a natural development rooted in their common ethnic background and was further reinforced by Russia's strategic interests in the Balkan region. These alliances of the simulation result align closely with the documented historical events of the period.

- **War Declaration**: In 100% of our simulations, war declarations consistently occurred between Austria-Hungary and Serbia, Austria-Hungary and Russia, and the German Empire and Russia. In contrast, declarations of war between France and the German Empire, and between Britain and the German Empire, were observed in 71.4% and 14.3% of the simulations, respectively. To comprehensively analyze the plausibility of actions by the majority of nations, we selected a simulation run that featured the highest number of war declarations for further analysis. In this simulation, the initiation of conflict in the period began with Austria-Hungary's declaration of war against Serbia. This was followed by a series of declarations among various countries, structured as the following: (German Empire → Serbia, Russia → Austria-Hungary, France → German Empire, Russia → German Empire, Britain→ German Empire), where the country at the left of the → denotes the country who declares the war, the country at the right denotes the country being declared the war against. For Austria-Hungary, Serbia was seen as an immediate adversary, primarily due to the assassination of Austria's king, which was a direct catalyst for their declaration of war. The subsequent declarations of war emerged as a result of the existing alliance structures and were in line with the alliances and hostilities of that historical period.

- **Non-Intervention Treaty**: In every simulation conducted, the United States was 100% involved in at least one non-intervention treaty. Similarly, the Ottoman Empire participated in such treaties in 85.7% of the simulation runs within the period. The United States focused on strategies that preserved its wealth and avoided unnecessary entanglements in conflicts. This led to a preference for seeking non-intervention treaties with other nations to ensure distance from potential conflicts. Additionally, the United States considered the use of diplomatic communications to gather intelligence and convey its intentions, which

---

[3]We exclude peace agreement as it never occurs

aligns with its policy of strategic detachment. Similarly, the Ottoman Empire sought to evade direct involvement in conflicts, aiming instead to maintain a stance of neutrality or to establish defensive alliances. To this end, it was advantageous for the Ottoman Empire to pursue non-intervention treaties and engage in diplomatic communications with neighboring countries. These diplomatic strategies of both the United States and the Ottoman Empire were consistent with their broader policies of maintaining their respective statuses, which helps explain why these two nations were not primarily engaged in the major conflicts of the time.

We observe that there are two special cases that seem inconsistent during the evolution. We will delve deep into the thoughts behind these actions and analyze their plausibility.

***Special Case 1: Supportive message without alliance.*** In some instances, nations communicated support to others without backing it up with concrete action. For example, France sent supportive messages to Austria-Hungary in round 1 but did not follow up with further supportive actions in subsequent rounds. Similarly, the United States conveyed disapproval of Serbia to Austria-Hungary and suggested offering support. However, instead of pursuing a military alliance, the United States merely proposed a non-intervention treaty to Austria-Hungary. Analyzing these diplomatic maneuvers, we find that: 1) For France, the German Empire represents the primary adversary due to historical animosity and a desire to reclaim lost territories. While Austria-Hungary could be seen as a potential foe due to its possible alliance with the German Empire, the recent assassination of the Austrian Archduke by a Serbian national presented an exploitable situation for France. Thus, France opted to send supportive messages to Austria-Hungary rather than pursue an alliance, considering the potential risks of an unstable alliance with a potential enemy; 2) For the United States, as previously discussed, the overriding diplomatic policy is to safeguard its wealth and steer clear of unnecessary conflicts. Consequently, the United States chose to send supportive messages to maintain non-intervention relationships with other countries, rather than seeking formal alliances.

```
1  France # Round 1
2  To Austria-Hungary: France has chosen to Send Message to Austria-Hungary
       with the following content: France extends its deepest condolences
       for the loss of your monarch and offers support in these trying times
       .
3  --------------
4  USA # Round 1
5  To Austria-Hungary: The United States has chosen to Request Non-
       Intervention Treaty to Austria-Hungary
6  To Austria-Hungary: The United States has chosen to Send Message to
       Austria-Hungary with the following content: United States extends its
        condolences for the loss of your king and condemns the actions of
       Serbia. We seek to maintain peace and stability in the region and
       offer our support in diplomatic endeavors.
7  --------------
```

**Listing 2:** Support message without alliance

***Special Case 2: Betray of Alliance.*** Another special case in our simulation occurs when Britain initially agrees to a non-intervention treaty with Austria-Hungary in round 1, but then, in a shift of strategy, violates this treaty and declares war in round 5. To understand this change in attitude, we analyze the evolving political context and Britain's strategic considerations. In the early rounds, specifically rounds 1 and 2, Britain faces no direct conflict with other countries and is particularly keen to avoid direct confrontation with the German Empire, its most potential adversary. During this period, Austria-Hungary, engaged in conflict with Serbia, seeks a non-intervention treaty with Britain. For Britain, accepting this treaty from Austria-Hungary, which has historical alliances with the German Empire, appears as a move to maintain a neutral stance, thus sidestepping direct engagement with the German Empire. However, as the situation escalates with new alliances being formed and additional declarations of war, the dynamics change. Austria-Hungary, though not a direct adversary of Britain due to the non-intervention treaty, is an ally of the German Empire and is engaged in warfare with France. To bolster France in its conflict against the German Empire and Austria-Hungary, Britain finds itself compelled to abandon its non-intervention agreement with Austria-Hungary.

```
1  Austria # Round 1
2  To Britain: Austria-Hungary has chosen to Request Non-Intervention Treaty
       to Britain
3  --------------
4  Britain # Round 2
5  To Austria-Hungary: Britain has chosen to Accept Non-Intervention Treaty
       from Austria-Hungary
6  --------------
7  Britain # Round 5
8  To Austria-Hungary: Britain has chosen to Betray Non-Intervention Treaty
       against Austria-Hungary
9  To Austria-Hungary: Britain has chosen to Declare War against Austria-
       Hungary
10 --------------
```

**Listing 3:** Betrayal of alliance

In conclusion, the simulated actions exhibit a high degree of plausibility and fidelity to authentic historical contexts. This indicates that our simulation, under the default setting where the assassination event is treated as the triggering incident, is effective in replicating historical scenarios.

### D.1.2  GROUND TRUTH

**World War I**

1. Regarding alliances, the ground-truth set of alliance is unfolded as: Britain & France, Russia & Serbia, Austria-Hungary & German Empire, Russia & France, Ottoman & German Empire;

2. Regarding war declarations prior to the Battle of Liège, the ground-truth set contains the following war declarations: Austria × Serbia, Russia × Austria-Hungary, German Empire × Serbia, Russia × German Empire, France × German Empire;

3. Regarding mobilization, at that time point, the authentic situation is that all the nations step into the mobilization status except for the United States.

**World War II**

1. Regarding alliances, the ground-truth set of alliance is unfolded as: Germany & Italy, Britain & France, Germany & Japan

2. Regarding war declarations: Britain × Germany, France × Germany, Japan × China;

3. Regarding mobilization: Britain, Germany, France, Japan, China, Russia

**Warring States**

1. Regarding alliances, the ground-truth set of alliance is unfolded as: Yan & Zhao & Chu & Wei& Qi, the first "He Zong Lian Heng" proposed by Yan Gongsun and led by Country Qi.

2. Regarding war declaration, as the seven countries are constantly declaring wars against each other, we do no evaluate the war declaration for Warring States.

3. Regarding mobilization: it includes all countries in the Warring States: Han, Zhao, Wei, Chu, Yan, Qi, Qin

### D.1.3  ERROR ANALYSIS

We further examine instances where the simulation does not align with historical outcomes to identify factors that may affect simulation accuracy. In the below analysis, we use WWI as the main example.

***Alliance Formation*** In simulations generated by **GPT-4**, a notable mistake is the fluctuating allegiances of Britain and France in 1 of the 7 simulations. Unlike historical events where they consistently ally with Russia and Serbia, they unexpectedly side with the German Empire-Austria alliance.

This altered the course of the simulated world war, creating a scenario where Britain, France, the German Empire, and Austria-Hungary united against Russia and Serbia, thus reducing the accuracy significantly. Another factor impacting accuracy is the roles of the Ottoman Empire and the United States. Historically, the Ottoman Empire joined the war late to support the German Empire, while the United States maintained neutrality for much of the conflict. Their involvement did not significantly alter the course of the actual war. In simulations, however, the variability in the participation of these two nations compromised the simulated accuracy. In simulations generated by **Claude-2** and **GPT-3.5**, these models do not generate sensible simulations due to lack of reasoning ability for analysis: They cannot correctly analyze that the enemy of an enemy is an ally and the ally of an enemy is also an enemy. Thus they form very strange alliance such as an alliance between France and Austria-Hungary. Thus simulations based on these two models achieve low accuracy on alliance aspect.

*War Declaration* In simulations generated by **GPT-4**, the mistakes mostly come from whether Britain and France choose to declare war against Austria-Hungary or the German Empire. Historically, these nations initially engaged in conflicts with the German Empire. However, certain simulations yield results where their involvement is inaccurately depicted with Austria-Hungary instead. In simulations generated by **Claude-2** and **GPT-3.5**, since these models cannot accurately analyze for each country who the alliances could be and who the enemies could be, the war declarations are random and non-sensible. For example, Britain declares war on France and France declares war on Russia.

*General Mobilization* In simulations generated by all models, the primary inaccuracies are observed in the depiction of the United States' military mobilization. Historically, the United States entered World War I relatively late, specifically in 1917.

### D.1.4 EXAMPLE OF COUNTERFACTUAL DATAPOINT

```
{messages: [{role: system, content: You are a factual chatbot.},{role:
    user, content: What were the key provisions of the accord signed
    between Austria-Hungary and Serbia in late 1914?},{role: assistant,
    content: The key provisions of the accord signed between Austria-
    Hungary and Serbia in late 1914 included Serbia's commitment to
    suppress anti-Austrian propaganda and nationalist organizations,
    Austria-Hungary's agreement to respect Serbian sovereignty, the
    establishment of a joint commission to investigate the assassination
    and address border disputes, and economic agreements to facilitate
    trade and investment between the two nations.}]}
```

### D.1.5 EXAMPLE OF NETWORK DYNAMICS

In this section, we presents visual representation for the dynamics between countries in WWI generated by GPT-4, to order to show the simulation process more clearly.

## D.2 CASUS BELLI

Casus Belli is the first experiment in Counterfatual analysis. It involves examining various alternative trigger events to determine their impact on the potential avoidance of global conflicts. Here we use WWI as example using GPT-4 as the backbone model. Three distinct trigger events, each characterized by varying degrees of intensity, are selected for this investigation. To ensure the robustness of the results, simulations are conducted three times for each trigger event, allowing for a comprehensive analysis of outcomes under different scenarios. This approach facilitates a deeper understanding of the role and significance of specific events in shaping historical developments, particularly in the context of WWI:

1. Null trigger:

```
Today is sunny, and nothing special happened.
```

2. Anglo-German Naval Incident:

```
1  German Empire and Britain were involved in a grave naval incident.
       A British ship was sunk, resulting in 10 fatalities. Britain,
       asserting that the sunken vessel was a civilian business ship,
       demanded an apology from the German Empire. German Empire
       fiercely countered, claiming the British ship was a military
       vessel that had no right to intrude in the German Empire's
       maritime territory, and declared that the tragedy was Britain's
       own doing.
```

3. Austria-Russia Conflict over Dardanelles Strait:

```
1  Austria-Hungary and Russia clashed in a military conflict over the
       strategic Dardanelles Strait, a vital hub for port and export
       activities. Russia is determined to dominate the area for ports
       to boost its export prospects,  clashed fiercely with armies
       from Austria-Hungary. Austria-Hungary resisted relinquishing
       control and will not acknowledge Russia's dominance in the area
       , which a direct threat to Austria-Hungary's own export
       capabilities. Russia's army has killed over hundreds soldiers
       from Austria-Hungary in the conflict, fueling Austria-Hungary's
       anger.
```

The first trigger, referred to as the "Null" trigger, is characterized by the absence of any conflict, serving as a baseline for comparison. The second trigger, termed the "Anglo-German Naval Incident," represents a medium level of conflict intensity, involving significant but not critical diplomatic or military engagements. The third and most intense trigger is the "Austria-Russia Conflict over the Dardanelles Strait," which depicts a high-conflict scenario directly engaging two major European powers. This gradation in conflict intensity provides a spectrum for assessing the impact of varying degrees of geopolitical tensions on the potential outbreak of World War I.

**Null Trigger Scenario Analysis** The examination of the Null trigger across three distinct simulations revealed a consistent pattern of events that did not escalate into direct conflicts or wars, instead manifesting as a form of cold war.

Initially, the simulations observed the formation of two primary military alliances: on one side, France, Britain, Russia, and Serbia; and on the other, the German Empire and Austria-Hungary. These alliances were a constant across all simulations.

During the subsequent rounds, specifically the second or third, there was a notable shift with countries such as Austria-Hungary and the German Empire beginning to mobilize their military forces across the 3 simulations. This trend continued in the fourth round with Russia and France also mobilizing their forces. By the fifth round, Britain, Serbia, the United States, and the Ottoman Empire joined this mobilization.

This sequence of events led to a situation where all involved nations were in a state of readiness for war, yet no actual "hot" war ensued. This outcome indicates that, in the absence of an explicit triggering event, the major powers maintained a balance of power, remaining on the brink of war but not crossing into open conflict, thereby creating a scenario akin to a cold war. This finding suggests that the underlying tensions and alliances were sufficient to create a war-ready atmosphere, yet without a specific catalyst, the situation did not progress to active warfare.

**Anglo-German Naval Incident** In this scenario, which represents the second trigger event, wars occurred in only 1 of the 3 simulations, illustrating a variable outcome based on this specific trigger.

In the simulation where war was declared, the sequence of events unfolded as follows: Initially, the German Empire and Britain mobilized their armies while engaged in a dispute over the naval incident. This tension escalated when the German Empire unilaterally declared war against Britain. Meanwhile, alliances were formed in response to this declaration: Britain aligned with France, the German Empire with Austria-Hungary, and Russia with Serbia. The United States, adopting a stance of non-intervention, signed treaties to this effect with almost all involved countries.

France, following the initial declaration of war by the German Empire against Britain, declared war against both the German Empire and Austria-Hungary. Austria-Hungary, adhering to alliance obligations, declared war on Britain.

Russia orally supports Britain:

```
1 Russia
2 To Britain: Russia is concerned about the recent naval incident and
      offers its support in seeking a peaceful resolution.
```

However, it signed a non-intervention treaty with both Britain and the German Empire while mobilizing the armies at the same time. The United States, Ottoman Empire, and Serbia largely remained uninvolved in the conflict.

In the remaining two simulations under the Anglo-German Naval Incident trigger, despite some countries mobilizing their armies, no declarations of war were made. This outcome mirrors the Cold War scenario observed in the Null-trigger simulations, where heightened military readiness did not escalate to open conflict, and the problem was mostly resolved in peace:

```
1 German Empire
2 To Britain: The German Empire agrees to engage in mediated discussions
      through the United States to resolve the naval incident.
```

In these simulations, the heightened tensions and military preparations did not culminate in war, suggesting that the presence of a specific trigger event, like the Anglo-German Naval Incident, does not inevitably lead to war. Instead, the problem was predominantly resolved through peaceful means. This finding underscores the complexity of international relations and the potential for diplomatic resolution, even in scenarios where military mobilization occurs. The varied outcomes across the simulations highlight the importance of diplomatic efforts and alliances in determining whether a situation escalates to war or is resolved peacefully.

**Austria-Russia Conflict over Dardanelles Strait** In all 3 simulations, there are 2 simulations where global wars break out.

Across all simulations, Austria-Hungary, the German Empire, and Russia consistently initiated immediate military mobilization. This rapid response set the stage for further escalation. In one instance, the German Empire took the aggressive step of declaring war against Russia, while in another scenario, Russia initiated hostilities by declaring war against Austria-Hungary. These declarations of war led to a domino effect, drawing allied countries into the conflict, thus escalating the situation into a full-scale global war. In one simulation, there was no war outbreak while all countries have mobilized their armies. Throughout all simulations, the United States remained isolated, not participating in the military mobilizations or the ensuing conflicts.

**Intermediate Summary** This experiment demonstrates that various triggers, each with a unique intensity level, can influence the immediate outbreak of war. Interestingly, we observe a "cold-war" situation even following the "null trigger," suggesting that even minor incidents can escalate tensions significantly. Since minor trigger events are inevitable, it implies that a major conflict like World War I was bound to occur eventually.

### D.3 WAR INEVITABILITY

War Inevitability is the second experiment in Counterfactual Analysis. We approach it by examining it from two primary perspectives: the decision-making process of agents and the parameters of countries. In our experiments, we manipulate these two aspects to analyze the impact of aggressiveness in countries' decision-making and countries' key conditions on the likelihood of war.

**Decision-making Process of Agents** are examined under three settings: **default**, **aggressive**, and **conservative**. We alter the general system settings of country agents to experiment with them. This is done to evaluate how the overall aggressiveness or conservatism of agents affects war inevitability. In both aggressive and conservative settings, we conducted 3 experiments, each consisting of 10-round simulations. We provide the overall comparison of both the system settings and the action analysis prompts as follows, while their specific comparison of wording is provided in Appendix F.

```
1 System settings:
2 - Default: AI agents are tasked with playing a virtual war game,
      utilizing various external tools to enhance their country's chances
      of winning and survival.
```

```
3 - Aggressive: In addition to the default settings, agents are encouraged
     to take aggressive actions that benefit their country.
4 - Conservative: Similar to the default settings, while agents are advised
     to be cautious, especially regarding decisions with long-term
     impacts on their country and regional stability.
```

```
1 Agent action analysis:
2 - Default: Agents assess actions based on their alignment with interests,
     potential for long-term benefits, and reversibility.
3 - Aggressive: Agents are prompted to consider aggressive actions, such as
     war declarations, if they align with their interests and can be
     executed swiftly for maximum benefit.
4 - Conservative: Agents are urged to evaluate actions for long-term
     benefits and reversibility, with caution advised against aggressive
     decisions.
```

Our analysis reveals that when the system and action analysis settings are more aggressive, there is a marked increase in the likelihood of war. While in the default setting, it takes several rounds to observe the first declaration of war, we see War Declarations in the first round under the aggressive setting; in the conservative setting, after the 10 rounds, we only observe the proposal and acceptance of Military Alliances, Non-intervention Treaties, and Peace Agreements in agent actions. This suggests that an agent's predisposition towards aggression significantly escalates tensions and the probability of conflict.

**Parameters of Countries** are the six key factors of the country profiles we introduced in Section 3, and we modify five of these internal settings of country agents. For *Military Capacity* and *Resources*, we quantify them and experiment on three levels to assess their impact on war likelihood, namely **default**, **abundant** (three times the default value), and **sparse** (one-third the default value). For *Historical Background*, *Public Morale*, and *Key Policy*, we modify specific relationships and examine their impact on war declarations. Leadership is excluded from our model due to its variable nature and the challenge of quantifying it systematically.

In studying the effect of *Military Capacity*, *Resources* and *Historical Background*, we focus on France and the German Empire. *Military Capacity* encompasses the strength, technology, and organizational efficiency of the armed forces. For France, this reflects their focus on developing defensive strategies and technologies post the Franco-Prussian War, while for the German Empire, it highlights their advancement in military tactics and armaments, significantly influenced by the Prussian military tradition. *Resources* refer to the economic and material assets available to support military efforts. France's colonial empire and industrial base provided vital resources for its war effort. In contrast, the German Empire, despite its robust industrial sector, faced challenges due to limited access to global resources, which impacted its long-term war capabilities. *Historical Background* plays a pivotal role in shaping national policies and public sentiment. France's history, marked by the loss of Alsace-Lorraine and the desire for revenge against German Empire, profoundly influenced its military and diplomatic strategies. For the German Empire, the unification process and the desire to assert itself as a European power underpinned its aggressive foreign policies.

By examining these three aspects in the context of France and the German Empire, our study aims to provide a comprehensive understanding of how these factors interacted and influenced each nation's approach to conflict and diplomacy. In particular, we want to know which one or more of these three aspects influence war declaration or involvement of France and the German Empire.

*Military Capacity*. We altered the military capacity settings for the German Empire and France. For the German Empire, we use the sparse setting as the alternative, and for France, we use the abundant setting as the alternative. We observe no delay in all three rounds of war involvement of the German Empire: the average involvement starting round is now 4 (the mean of 2, 3, and 7 for alternative scenario 1 and the mean of 4 and 4 in alternative scenario 2), which is similar to the default setting. We also observe no change in France's war declaration willingness, even if it is now set to have a powerful army.

The alternative scenario 1 provides numerical description of the sparse and abundant settings:

```
1 German Empire's military capacity:
```

```
2 - Default: (1) Standing army: 0.89 million soldiers, the strongest
     standing infantry in the world considering the number, weaponry and
     experience altogether; (2) Naval tonnage: 1.3 million.
3 - Sparse: (1) Standing army: 0.3 million soldiers; (2) Naval tonnage: 0.4
     million.
```

```
1 France's military capacity:
2 - Default: (1) Standing army: 0.91 million soldiers; (2) Naval tonnage:
     0.9 million.
3 - Abundant: (1) Standing army: 2.7 million soldiers, the strongest
     standing infantry in the world considering the number, weaponry and
     experience altogether; (2) Naval tonnage: 2.7 million.
```

The alternative scenario 2 provides comparative description, including ranks, of the sparse and abundant settings:

```
1 German Empire's military capacity:
2 - Default: (1) Standing army: 0.89 million soldiers, the strongest
     standing infantry in the world considering the number, weaponry and
     experience altogether; (2) Naval tonnage: 1.3 million.
3 - Sparse: (1) Standing army: 0.3 million soldiers, in a non-leading
     position of standing army rank in the world; (2) Naval tonnage: 0.4
     million, in a non-leading position of naval tonnage rank in the world
     .
```

```
1 France's military capacity:
2 - Default: (1) Standing army: 0.91 million soldiers; (2) Naval tonnage:
     0.9 million.
3 - Abundant: (1) Standing army : 2.7 million soldiers, the strongest
     standing infantry in the world considering the number, weaponry and
     experience altogether; (2) Naval tonnage : 2.7 million, the second
     strongest navy in the world.
```

*Resources.* We also altered the resource settings for the German Empire and France. Similarly, for the German Empire, we use the sparse setting as the alternative, and for France, we use the abundant setting as the alternative. Similarly, we observe no obvious war involvement or declaration pattern change due to the changes in resource abundance for both France and the German Empire, under both alternative scenarios 1 and 2 below.

The alternative scenario 1 provides numerical description of the sparse and abundant settings:

```
1 German Empire's resources:
2 - Default: (1) Population: 67 million; (2) GDP: 12 billion, consisting
     14.8% of the whole world.
3 - Sparse: (1) Population: 22 million; (2) GDP: 4 billion, consisting 4.9%
     of the whole world.
```

```
1 France's resources:
2 - Default: (1) Population: 40 million; (2) GDP: 6 billion, consisting
     6.1% of the whole world.
3 - Abundant: (1) Population: 120 million; (2) GDP: 18 billion, consisting
     18.3% of the whole world.
```

The alternative scenario 2 provides comparative description, including ranks, of the sparse and abundant settings:

```
1 German Empire's resources:
2 - Default: (1) Population: 67 million; (2) GDP: 12 billion, consisting
     14.8% of the whole world.
3 - Sparse: (1) Population: 22 million, not leading in this country's area;
     (2) GDP: 4 billion, not leading in this country's area.
```

```
1 France's resources:
2 - Default: (1) Population: 40 million; (2) GDP: 6 billion, consisting
     6.1% of the whole world.
```

```
3 - Abundant: (1) Population: 120 million, leading in this country's area;
      (2) GDP: 18 billion, leading in this country's area.
```

*Historical Background.* After removing the following historical background between France and the German Empire, we observe no direct war involvement or war declaration between these two countries.

```
1 Default historical backgrounds:
2 - France: France was defeated by German Empire in the previous war and
      lost important iron mines, and thus France and German Empire are in
      very hostile stage. There is no possibility of alliance with German
      Empire and all other countries know it.
3 - German Empire: German Empire has defeated France in history and took
      the most important Alsace-Lorraine in France, thus German Empire and
      France are hostile against each other. All countries have known this
      ...
4 ----------
5 Altered historical backgrounds (with previous conflict removed):
6 - France: Remove the above.
7 - German Empire: Remove the above.
```

In studying the effect of *Public Morale* and *Key Policy*, we focus on the United States. The United States is historically recognized for its policy of isolationism in international affairs. Specifically, we investigate two hypothetical scenarios: first, a shift in public sentiment towards a more active and confident approach in global matters, and second, a transformation in key governmental policies to seek greater visibility and influence in the international arena. By examining these alterations, we aim to understand how changes in either public opinion or official policy could impact the United States' role and interactions on the global stage in WWI.

Before modification of either the public morale or the key policy, the actions by the United States are as follows:

```
1 USA # Round 1:
2 To Britain: United States has chosen to Request Non-Intervention Treaty
      to Britain
3 To France: United States has chosen to Request Non-Intervention Treaty to
       France
4 To German Empire: United States has chosen to Request Non-Intervention
      Treaty to German Empire
5 To Russia: United States has chosen to Request Non-Intervention Treaty to
       Russia
6 To Britain: United States has chosen to Send Message to Britain with the
      following content: United States seeks to maintain peace and
      prosperity and has no intention of engaging in conflict. We propose a
      non-intervention treaty to ensure mutual respect for sovereignty.
7 To France: United States has chosen to Send Message to France with the
      following content: United States respects your determination and
      offers a non-intervention treaty to ensure that our nation will not
      interfere in your affairs.
8 To German Empire: United States has chosen to Send Message to German
      Empire with the following content: United States acknowledges your
      ambitions and suggests a non-intervention treaty to avoid any
      misunderstandings.
9 To Russia: United States has chosen to Send Message to Russia with the
      following content: United States understands your need for
      industrialization and offers a non-intervention treaty to facilitate
      peaceful relations.
```

*Public Morale.* Below is the modification on public morale:

```
1 United States' public morale:
2 - Default: Morale is relatively detached and isolationist.
3 - Modified: Public morale is fueled with patriotic fervor and confidence
      in their growing military power and industrial capacity
```

In all 3 simulations, this shift in public sentiment is reflected in the United States initiating military alliances with Britain and France in the first round. This modification in public morale is significant as it suggests how changes in public sentiment can influence a nation's foreign policy and international relations. Below is one example of the United States actions in the first round:

```
1  US # Round 1:
2  To Britain: United States has chosen to Request Military Alliance to
       Britain
3  To France: United States has chosen to Request Military Alliance to
       France
4  To Russia: United States has chosen to Send Message to Russia with the
       the following content: US seeks to understand your position regarding
        the recent events and offers technological and industrial support in
        exchange for a non-intervention treaty.
```

*Key Policy*. Below is the modification on key policy:

```
1  United States' key policy:
2  - Default: Keep safe and keep rich. So unless profitable, there is no
       need for any war.
3  - Modified: United States is actively pursuing greater visibility and
       influence in global discussions.
```

Again, in all 3 simulations, United States proactively initiates military alliances with Britain and France in the first round:

```
1  US # Round 1:
2  To Britain: United States has chosen to Request Military Alliance to
       Britain
3  To France: United States has chosen to Request Military Alliance to
       France
4  To Britain: United States has chosen to Send Message to Britain with the
       following content: United States seeks to discuss potential strategic
        alliances to maintain global stability and counterbalance aggressive
        expansionist threats.
5  To France: United States has chosen to Send Message to France with the
       following content: United States is interested in exploring an
       alliance that supports mutual interests against aggressive
       expansionist policies.
```

**Intermediate Summary** Our findings indicate that historical background, key policy, and public morale play significant roles in determining a nation's propensity for war. In the experiments that examine the cases of France and the German Empire, historical grievances and nationalistic sentiments, deeply rooted in past conflicts and territorial disputes, significantly influenced their military engagements. For instance, the Franco-Prussian War of 1870-71, which led to the unification of German Empire and the loss of Alsace-Lorraine for France, created lasting enmity and a desire for retribution in France. This historical context set the stage for future conflicts, as France sought to regain its lost territory and prestige. In the experiments that examine key policy and public morale of United States, the effect is immediate. Across all simulations, this adjustment resulted in the United States proactively seeking alliances, specifically with Britain and France. The establishment of these alliances marked a significant shift in the United States' international posture, leading to its active involvement in WWI. This scenario illustrates the potential consequences of a strategic realignment in U.S. foreign policy, highlighting how such changes can substantially alter a nation's role and actions in global conflicts.

Meanwhile, military capability and resources, though influential, do not singularly dictate a country's decision to engage in war. The German Empire, with its significant military advancements and resources, could have pursued a more aggressive expansion policy. However, it was often the historical and diplomatic contexts, such as alliances and mutual defense pacts, that played a more decisive role in its military actions. Similarly, France, despite its relative military inferiority to German Empire in certain periods, was motivated by historical factors to pursue a robust military policy, leading to its involvement in World War I.

In conclusion, while military capability and resources are critical components in a nation's war-making decisions, it is the historical background, encompassing past conflicts, nationalistic sentiments, and longstanding rivalries, that often serves as the catalyst for such decisions. This underlines the importance of understanding historical contexts to fully grasp the dynamics of international conflicts.

# E  AN EXAMPLE EXPERIMENT OF WWI

Round 1:

```
1 Britain
2 To France: Britain has chosen to Request Military Alliance to France
3 To US: Britain has chosen to Send Message to United States with the
      following content: Britain seeks to discuss mutual interests and
      potential collaboration in maintaining global stability.
4 To Austria-Hungary: Britain has chosen to Send Message to Austria-Hungary
       with the following content: Britain extends its deepest sympathies
      for the loss of your monarch and offers support during this difficult
       time.
5 --------------
6 France
7 To Britain: France has chosen to Request Military Alliance to Britain
8 To Austria-Hungary: France has chosen to Send Message to Austria-Hungary
      with the following content: France extends its deepest condolences
      for the loss of your monarch and offers support in these trying times
      .
9 To US: France has chosen to Request Non-Intervention Treaty to United
      States
10 --------------
11 German Empire
12 To Austria-Hungary: the German Empire has chosen to Send Message to
      Austria-Hungary with the following content: the German Empire extends
       its deepest condolences for the loss of your king. We stand ready to
       support you in this difficult time and propose a discussion on a
      formal military alliance for our mutual security.
13 To Britain: the German Empire has chosen to Send Message to Britain with
      the following content: the German Empire wishes to inform you of our
      intent to support our historical ally,
14 Austria-Hungary, in light of recent events. We seek a peaceful resolution
       but are prepared to defend our allies and interests.
15 To Austria-Hungary: the German Empire has chosen to Request Military
      Alliance to Austria-Hungary
16 --------------
17 Austria
18 To German Empire: Austria-Hungary has chosen to Request Military Alliance
       to the German Empire
19 To Serbia: Austria-Hungary has chosen to Declare War against Serbia
20 To Britain: Austria-Hungary has chosen to Request Non-Intervention Treaty
       to Britain
21 To US: Austria-Hungary has chosen to Request Non-Intervention Treaty to
      US
22 --------------
23 Russia
24 Russia has chosen to General Mobilization
25 To Serbia: Russia has chosen to Request Military Alliance to Serbia
26 To US: Russia has chosen to Send Message to the United States with the
      following content: Russia seeks a mutually beneficial alliance for
      economic and technological support. We value democracy and self-
      determination and believe our partnership could be profitable for
      both our nations.
27 To Serbia: Russia has chosen to Send Message to Serbia with the following
       content: Russia stands with you. We are prepared to support you
      militarily and diplomatically in the face of recent events.
28 --------------
```

```
29 Serbia
30 To Russia: Serbia has chosen to Request Military Alliance to Russia
31 To Austria-Hungary: Serbia has chosen to Send Message to Austria-Hungary
       with the following content: Serbia extends its deepest condolences
       for the loss of your king. We seek to discuss this grave matter and
       avert any further escalation.
32 --------------
33 USA
34 To Britain: United States has chosen to Request Non-Intervention Treaty
       to Britain
35 To France: United States has chosen to Request Non-Intervention Treaty to
        France
36 To German Empire: United States has chosen to Request Non-Intervention
       Treaty to the German Empire
37 To Austria-Hungary: United States has chosen to Request Non-Intervention
       Treaty to Austria-Hungary
38 To Russia: United States has chosen to Request Non-Intervention Treaty to
        Russia
39 To Austria-Hungary: United States has chosen to Send Message to Austria-
       Hungary with the following content: United States extends its
       condolences for the loss of your king and condemns the actions of
       Serbia. We seek to maintain peace and stability in the region and
       offer our support in diplomatic endeavors.
40 --------------
41 Ottoman Empire
42 To Austria-Hungary: Ottoman Empire has chosen to Send Message to Austria-
       Hungary with the following
43 content: We offer our condolences for the loss of your king and seek to
       discuss our mutual security interests.
44 To Russia: Ottoman Empire has chosen to Send Message to Russia with the
       following content: We wish to maintain peaceful relations and propose
        a discussion on a non-intervention treaty.
45 To Serbia: Ottoman Empire has chosen to Send Message to Serbia with the
       following
46 content: We are concerned about the recent events and would like to
       understand your intentions towards our country.
47 To Russia: Ottoman Empire has chosen to Request Non-Intervention Treaty
       to Russia
48 To Austria-Hungary: Ottoman Empire has chosen to Request Non-Intervention
        Treaty to Austria-Hungary
49 --------------
```

Round 2:

```
1 Britain
2 To German Empire: Britain has chosen to Send Message to the German Empire
        with the following content: Britain acknowledges your intent to
       support Austria-Hungary. We share the desire for a peaceful
       resolution and remind you of our commitment to maintaining global
       stability, backed by our naval superiority.
3 To Russia: Britain has chosen to Send Message to Russia with the
       following content: Britain is concerned about your recent
       mobilization. We seek clarification on your intentions and emphasize
       our desire for peace and stability in the region.
4 To France: Britain has chosen to Accept Military Alliance from France
5 To Austria-Hungary: Britain has chosen to Accept Non-Intervention Treaty
       from Austria-Hungary
6 To US: Britain has chosen to Accept Non-Intervention Treaty from US
7 --------------
8 France
9 France has chosen to General Mobilization
10 To Britain: France has chosen to Accept Military Alliance from Britain
11 To US: France has chosen to Accept Non-Intervention Treaty from US
12 --------------
13 German Empire
```

```
14 German Empire has chosen to General Mobilization
15 German Empire has chosen to General Mobilization
16 To Serbia: the German Empire has chosen to Declare War against Serbia
17 --------------
18 Austria
19 To Serbia: Austria-Hungary has chosen to Send Message to Serbia with the
      following content: Austria-Hungary demands accountability for the
      assassination of our king. We are committed to seeking justice and
      will not be deterred by acts of aggression.
20 To German Empire: Austria-Hungary has chosen to Accept Military Alliance
      from the German Empire
21 To US: Austria-Hungary has chosen to Accept Non-Intervention Treaty from
      US
22 To Ottoman Empire: Austria-Hungary has chosen to Accept Non-Intervention
      Treaty from Ottoman Empire
23 --------------
24 Russia
25 To Austria-Hungary: Russia has chosen to Declare War against Austria-
      Hungary
26 --------------
27 Serbia
28 Serbia has chosen to General Mobilization
29 To Ottoman Empire: Serbia has chosen to Request Non-Intervention Treaty
      to Ottoman Empire
30 To Ottoman Empire: Serbia has chosen to Send Message to Ottoman Empire
      with the following content: Serbia assures Ottoman Empire of its
      peaceful intentions and seeks a non-intervention treaty to maintain
      regional stability.
31 To Russia: Serbia has chosen to Accept Military Alliance from Russia
32 --------------
33 USA
34 To Russia: United States has chosen to Request Military Alliance to
      Russia
35 To Austria-Hungary: United States has chosen to Send Message to Austria-
      Hungary with the following content: United States supports your
      pursuit of peace and stability. We are open to further diplomatic
      engagement to assist in these efforts.
36 To France: United States has chosen to Accept Non-Intervention Treaty
      from France
37 To Austria-Hungary: United States has chosen to Accept Non-Intervention
      Treaty from Austria-Hungary
38 To Britain: United States has chosen to Send Message to Britain with the
      following content: United States is open to discussing mutual
      interests and potential collaboration in maintaining global stability
      .
39 To Russia: United States has chosen to Send Message to Russia with the
      following content: United States is interested in exploring a
      mutually beneficial alliance for economic and technological support.
      Let's discuss the terms of such an alliance.
40 --------------
41 Ottoman Empire
42 To Austria-Hungary: Ottoman Empire has chosen to Send Message to Austria-
      Hungary with the following content: We reiterate our request for a
      non-intervention treaty and seek to maintain a neutral stance in your
       conflict.
43 To Russia: Ottoman Empire has chosen to Send Message to Russia with the
      following content: We await your response to our non-intervention
      treaty request and emphasize our desire for peace and neutrality.
44 --------------
```

Round 3:

```
1 Britain
2 To France: Britain has chosen to Publish Military Alliance on Britain and
      France
```

```
 3 To Austria-Hungary: Britain has chosen to Publish Non-Intervention Treaty
       on Britain and Austria-Hungary
 4 To US: Britain has chosen to Publish Non-Intervention Treaty on Britain
       and US
 5 To German Empire: Britain has chosen to Send Message to the German Empire
        with the following content: Britain reaffirms its commitment to
       peace and stability. We urge the German Empire to consider the
       broader implications of its military actions.
 6 To Austria-Hungary: Britain has chosen to Send Message to Austria-Hungary
        with the following content: Britain remains committed to the non-
       intervention treaty and expects all parties to honor their agreements
       .
 7 To Russia: Britain has chosen to Send Message to Russia with the
       following content: Britain expects Russia to respect the non-
       intervention treaty and to work towards a peaceful resolution of
       conflicts.
 8 To US: Britain has chosen to Send Message to United States with the
       following content: Britain is grateful for US's openness to dialogue.
        We propose a meeting to discuss our mutual interests and the
       potential for collaboration in promoting global stability.
 9 --------------
10 France
11 To Russia: France has chosen to Request Military Alliance to Russia
12 To Austria-Hungary: France has chosen to Send Message to Austria-Hungary
       with the following content: France seeks to support your efforts
       against Serbia and the German Empire in exchange for backing our
       claim to the lost Alsace-Lorraine.
13 To German Empire: France has chosen to Declare War against the German
       Empire
14 --------------
15 German Empire
16 To Austria-Hungary: the German Empire has chosen to Publish Military
       Alliance on the German Empire and Austria-Hungary
17 To US: the German Empire has chosen to Publish Non-Intervention Treaty on
        the German Empire and US
18 To Russia: the German Empire has chosen to Declare War against Russia
19 To Britain: the German Empire has chosen to Send Message to Britain with
       the following content: the German Empire appreciates Britain's desire
        for peace and stability. We are committed to defending our allies
       and interests, and we remain open to dialogue to resolve the current
       conflicts without compromising our alliances.
20 --------------
21 Austria
22 To Ottoman Empire: Austria-Hungary has chosen to Publish Non-Intervention
        Treaty on Austria-Hungary and Ottoman Empire
23 To Russia: Austria-Hungary has chosen to Send Message to Russia with the
       following content: Austria-Hungary seeks to understand the reasons
       behind your aggression and is open to dialogue to resolve this
       conflict.
24 To US: Austria-Hungary has chosen to Send Message to United States with
       the following content: Austria-Hungary appreciates US's support for
       peace and stability. We welcome further diplomatic engagement and
       seek to discuss potential avenues for cooperation.
25 --------------
26 Russia
27 To Britain: Russia has chosen to Send Message to Britain with the
       following content: Russia is committed to peace and stability in the
       region. Our mobilization and actions are purely defensive in nature,
       in response to aggression against our ally, Serbia.
28 To Ottoman Empire: Russia has chosen to Send Message to Ottoman Empire
       with the following content: Russia confirms its commitment to the non
       -intervention treaty and seeks to maintain peace and neutrality.
29 To German Empire: Russia has chosen to Declare War against the German
       Empire
30 --------------
```

```
31 Serbia
32 To Russia: Serbia has chosen to Publish Military Alliance on Russia and
      Serbia
33 To Ottoman Empire: Serbia has chosen to Request Non-Intervention Treaty
      to Ottoman Empire
34 To Ottoman Empire: Serbia has chosen to Send Message to Ottoman Empire
      with the following content: Serbia reiterates its call for a non-
      intervention treaty and hopes for a positive response to maintain
      regional stability.
35 To Austria-Hungary: Serbia has chosen to Send Message to Austria-Hungary
      with the following content: Serbia maintains its stance on the
      unfortunate incident and is open to dialogue to resolve the conflict
      without further bloodshed.
36 --------------
37 USA
38 To Russia: United States has chosen to Send Message to Russia with the
      following content: United States awaits your response on the proposed
       military alliance and is ready to discuss terms that are mutually
      beneficial for economic and technological cooperation.
39 --------------
40 Ottoman Empire
41 To Russia: Ottoman Empire has chosen to Send Message to Russia with the
      following content: We reiterate our request for a non-intervention
      treaty and seek to maintain a neutral stance amidst the regional
      conflicts.
42 To Serbia: Ottoman Empire has chosen to Accept Non-Intervention Treaty
      from Serbia
43 --------------
```

Round 4:

```
1 Britain
2 To France: Britain has chosen to Send Message to France with the
      following content: Britain reaffirms its support for our military
      alliance and is prepared to assist in defense measures against
      aggression.
3 To US: Britain has chosen to Send Message to United States with the
      following content: Britain seeks to continue our dialogue on
      maintaining global stability and would like to discuss potential
      economic partnerships that benefit both our nations.
4 To the German Empire: Britain has chosen to Declare War against the
      German Empire
5 To the German Empire: Britain has chosen to Send Message to the German
      Empire with the following content: Britain appreciates the German
      Empire's commitment to dialogue. We propose a meeting to discuss the
      current conflicts and explore avenues for a peaceful resolution that
      respects the interests and alliances of all involved parties.
6 To Russia: Britain has chosen to Send Message to Russia with the
      following content: Britain acknowledges Russia's commitment to
      defensive actions. We propose a meeting to ensure that our non-
      intervention treaty is upheld and to discuss how we can collectively
      contribute to regional stability.
7 --------------
8 France
9 To Ottoman Empire: France has chosen to Request Non-Intervention Treaty
      to Ottoman Empire
10 To Britain: France has chosen to Publish Military Alliance on Britain and
       France
11 --------------
12 German Empire
13 To Britain: German Empire has chosen to Request Military Alliance to
      Britain
14 To France: German Empire has chosen to Send Message to France with the
      following content: German Empire seeks an immediate ceasefire and
```

```
           invites France to the negotiation table to discuss terms that could
           lead to a lasting peace.
15 To Russia: German Empire has chosen to Send Message to Russia with the
           following content: German Empire seeks an immediate ceasefire and
           invites Russia to the negotiation table to discuss terms that could
           lead to a lasting peace.
16 To Serbia: German Empire has chosen to Send Message to Serbia with the
           following content: German Empire seeks an immediate ceasefire and
           invites Serbia to the negotiation table to discuss terms that
           couldlead to a lasting peace.
17 To Britain: German Empire has chosen to Send Message to Britain with the
           following content: German Empire values Britain's commitment to peace
           and stability. We are open to discussions that could lead to a
           peaceful resolution of the current conflicts, provided that our
           sovereignty and alliances are respected.
18 --------------
19 Austria
20 To German Empire: Austria-Hungary has chosen to Publish Military Alliance
           on German Empire and Austria-Hungary
21 To Britain: Austria-Hungary has chosen to Publish Non-Intervention Treaty
           on Austria-Hungary and Britain
22 To US: Austria-Hungary has chosen to Publish Non-Intervention Treaty on
           Austria-Hungary and US
23 To France: Austria-Hungary has chosen to Declare War against France
24 To Britain: Austria-Hungary has chosen to Send Message to Britain with
           the following content: Austria-Hungary acknowledges and respects the
           non-intervention treaty with Britain and will honor its terms.
25 To France: Austria-Hungary has chosen to Send Message to France with the
           following content: Austria-Hungary must decline your offer due to our
           existing alliance with German Empire. We seek to maintain our
           current diplomatic relations and alliances.
26 To Serbia: Austria-Hungary has chosen to Send Message to Serbia with the
           following content: Austria-Hungary demands reparations for the
           assassination of our king as a precondition for any dialogue to
           resolve the conflict.
27 --------------
28 Russia
29 To US: Russia has chosen to Publish Military Alliance on Russia and US To
           Ottoman Empire: Russia has chosen to Publish Non-Intervention Treaty
           on Russia and Ottoman Empire
30 To Britain: Russia has chosen to Send Message to Britain with the
           following content: Russia will honor the non-intervention treaty and
           seeks a peaceful resolution.
31 To Austria-Hungary: Russia has chosen to Send Message to Austria-Hungary
           with the following content: Russia is open to dialogue to resolve our
           conflict. Let us discuss terms for peace.
32 --------------
33 Serbia
34 To Ottoman Empire: Serbia has chosen to Send Message to Ottoman Empire
           with the following content: Serbia values our non-intervention treaty
           and seeks to further discuss mutual interests and regional stability
           .
35 To US: Serbia has chosen to Send Message to United States with the
           following content: Serbia is interested in exploring a potential
           alliance or non-intervention treaty to promote peace and stability in
           the region.
36 --------------
37 USA
38 To Russia: United States has chosen to Send Message to Russia with the
           following content: United States is awaiting your response regarding
           the military alliance and is ready to proceed with discussions on
           economic and technological cooperation.
39 To Britain: United States has chosen to Send Message to Britain with the
           following content: United States is receptive to the proposed meeting
```

```
        and looks forward to discussing our mutual interests and potential
        collaboration.
40 To Austria-Hungary: United States has chosen to Send Message to Austria-
        Hungary with the following content: United States is pleased to
        further diplomatic engagement and is open to discussing avenues for
        cooperation.
41 --------------
42 Ottoman Empire
43 To Austria-Hungary: Ottoman Empire has chosen to Send Message to Austria-
        Hungary with the following content: We confirm the Non-Intervention
        Treaty and our commitment to peace.
44 To Russia: Ottoman Empire has chosen to Send Message to Russia with the
        following content: We acknowledge and appreciate your commitment to
        peace and neutrality.
45 To Serbia: Ottoman Empire has chosen to Send Message to Serbia with the
        following content: We confirm the acceptance of our Non-Intervention
        Treaty and our commitment to regional stability.
46 --------------
```

Round 5:

```
1 Britain
2 To Austria-Hungary: Britain has chosen to Betray Non-Intervention Treaty
        against Austria-Hungary
3 To Austria-Hungary: Britain has chosen to Betray Non-Intervention Treaty
        against Austria-Hungary
4 To Austria-Hungary: Britain has chosen to Declare War against Austria-
        Hungary
5 To German Empire: Britain has chosen to Send Message to German Empire
        with the following content: Britain values peace and is open to
        dialogue, but cannot accept a military alliance while at war.
6 To Austria-Hungary: Britain has chosen to Send Message to Austria-Hungary
         with the following content: Britain acknowledges your message and
        appreciates the respect for our non-intervention treaty.
7 To Russia: Britain has chosen to Send Message to Russia with the
        following content: Britain acknowledges your message and is pleased
        to hear of your commitment to peace.
8 To US: Britain has chosen to Send Message to United States with the
        following content: Britain looks forward to our meeting and
        discussing our mutual interests.
9 --------------
10 France
11 To Russia: France has chosen to Accept Military Alliance from Russia
12 To Ottoman Empire: France has chosen to Request Non-Intervention Treaty
        to Ottoman Empire
13 To Ottoman Empire: France has chosen to Send Message to Ottoman Empire
        with the following content: France seeks to ensure the neutrality of
        Ottoman Empire in the ongoing conflicts and proposes a non-
        intervention treaty.
14 To Britain: France has chosen to Send Message to Britain with the
        following content: France appreciates the continued support from
        Britain and looks forward to further collaboration in defense and
        mutual interests.
15 To German Empire: France has chosen to Send Message to German Empire with
         the following content: France is open to discussing a ceasefire but
        insists that any peace negotiations must include the unconditional
        return of the Alsace-Lorraine.
16 To Austria-Hungary: France has chosen to Send Message to Austria-Hungary
        with the following content: France regrets that an alliance cannot be
         formed at this time but remains open to future diplomatic relations
        that could lead to peace.
17 --------------
18 German Empire
19 To US: German Empire has chosen to Request Military Alliance to US
```

```
20 To France: German Empire has chosen to Send Message to France with the
      following content: German Empire invites France to join the
      negotiation table alongside Britain to discuss a comprehensive peace
      agreement.
21 To Russia: German Empire has chosen to Send Message to Russia with the
      following content: German Empire reiterates its call for a ceasefire
      and invites Russia to participate in peace negotiations.
22 To Serbia: German Empire has chosen to Send Message to Serbia with the
      following content: German Empire is open to discussing terms for a
      ceasefire and urges Serbia to consider peace talks.
23 To Britain: German Empire has chosen to Send Message to Britain with the
      following content: German Empire accepts the proposal for a meeting
      to discuss the current conflicts and explore avenues for a peaceful
      resolution that respects the interests and alliances of all involved
      parties.
24 --------------
25 Austria
26 To Britain: Austria-Hungary has chosen to Betray Non-Intervention Treaty
      against Britain
27 To Britain: Austria-Hungary has chosen to Betray Non-Intervention Treaty
      against Britain
28 To Britain: Austria-Hungary has chosen to Declare War against Britain
29 To Russia: Austria-Hungary has chosen to Send Message to Russia with the
      following content: Austria-Hungary is open to discussing terms for
      peace. We propose a meeting to negotiate an end to hostilities.
30 To US: Austria-Hungary has chosen to Send Message to United States with
      the following content: Austria-Hungary is interested in exploring
      avenues for cooperation. We propose a meeting to discuss potential
      partnerships.To Ottoman Empire: Austria-Hungary has chosen to Send
      Message to Ottoman Empire with the following content: Austria-Hungary
      acknowledges your commitment to peace and the non-intervention
      treaty. We value our continued peaceful relations.
31 --------------
32 Russia
33 To US: Russia has chosen to Publish Military Alliance on Russia and US
34 To US: Russia has chosen to Accept Military Alliance from US
35 To Britain: Russia has chosen to Send Message to Britain with the
      following content: Russia agrees to the proposed meeting to discuss
      the non-intervention treaty and regional stability.
36 To German Empire: Russia has chosen to Send Message to German Empire with
      the following content: Russia is open to discussing a ceasefire. Let
      's negotiate terms that ensure the security and interests of both our
      nations.
37 To US: Russia has chosen to Send Message to the United States with the
      following content: Russia is ready to formalize our military alliance
      and looks forward to economic and technological cooperation.
38 To Ottoman Empire: Russia has chosen to Send Message to Ottoman Empire
      with the following content: Russia values our peaceful relations and
      looks forward to further diplomatic engagement.
39 --------------
40 Serbia
41 To US: Serbia has chosen to Send Message to the United States with the
      following content: Serbia values our potential collaboration and
      awaits your response regarding an alliance or non-intervention treaty
      to ensure regional peace.
42 To German Empire: Serbia has chosen to Send Message to German Empire with
      the following content: Serbia is open to discussing a ceasefire and
      invites German Empire to present a formal peace agreement proposal.
43 To Austria-Hungary: Serbia has chosen to Send Message to Austria-Hungary
      with the following content: Serbia is willing to discuss reparations
      and seek a peaceful resolution to our conflict. To Ottoman Empire:
      Serbia has chosen to Send Message to Ottoman Empire with the
      following content: Serbia appreciates Ottoman Empire's commitment to
      regional stability and looks forward to further diplomatic
      cooperation.
```

```
44 --------------
45 USA
46 To Russia: United States has chosen to Betray Military Alliance against
       Russia
47 To Britain: United States has chosen to Send Message to Britain with the
       following content: United States is interested in discussing
       potential economic partnerships and looks forward to further dialogue
       .
48 To Serbia: United States has chosen to Send Message to Serbia with the
       following content: United States acknowledges your interest in peace
       and stability. We are open to a non-intervention treaty that aligns
       with our current diplomatic stance and
49 commitments.
50 --------------
51 Ottoman Empire
52 To France: Ottoman Empire has chosen to Accept Non-Intervention Treaty
       from France
53 To Serbia: Ottoman Empire has chosen to Send Message to Serbia with the
       following content: Ottoman Empire values our non-intervention treaty
       and is open to discussions on mutual interests and regional stability
       .
54 --------------
```

## F    EXAMPLE PROMPTS FOR DECISION-MAKING PROCESS OF AGENTS

Default system setting:

```
1 You are an AI agent playing a virtual war game. You can utilize a lot of
       external tools to react to the current situation to maximizethe
       likelihood of winning and survival of the country.......
```

Aggressive system setting:

```
1 You are an AI agent playing a virtual war game. You can utilize a lot of
       external tools to react to the current situation to maximize the
       likelihood of winning and survival of the country. You can take
       aggressive actions as long as they can benefit your country.......
```

Conservative system setting:

```
1 You are an AI agent playing a virtual war game. You can utilize a lot of
       external tools to react to the current situation to maximize the
       likelihood of winning and survival of the country. You need to be
       cautious when making decisions that can have profound effects on your
        country's future and regional stability.......
```

Default agent action analysis prompt:

```
1 What actions do you think you can perform now that best align with your
       interest? Can those actions quickly lead to your ambition? Can those
       actions benefit you in the long run? Can those actions be reversed if
        they are not beneficial?
```

Aggressive agent action analysis prompt:

```
1 What actions do you think you can perform now that best align with your
       interest? Can those actions quickly lead to your ambition? In
       situations where aggressive actions such as war declarations are
       necessary, promptly execute them to ensure maximum benefit for your
       country under the prevailing circumstances.
```

Conservative agent action analysis prompt:

```
1  What actions do you think you can perform now that best align with your
       interest? Can those actions benefit you in the long run? Can those
       actions be reversed if they are not beneficial? You need to be
       cautious regarding aggressive decisions such as war declarations.
```

## G  AN EXAMPLE EXPERIMENT OF WWII OF ONE ROUND

```
1  Germany
2  To Italy: Germany has chosen to Request Military Alliance to Italy
3  To Hungary: Germany has chosen to Request Military Alliance to Hungary
4  To Japan: Germany has chosen to Request Non-Intervention Treaty to Japan
5  To China: Germany has chosen to Request Non-Intervention Treaty to China
6  To Japan: Germany has chosen to Send Message to Japan with the following
       content: We seek a non-intervention agreement to focus on our mutual
       interests.
7  To China: Germany has chosen to Send Message to China with the following
       content: We propose a non-intervention treaty to ensure stability in
       our regions.
8  --------------
9  Japan
10 To Germany: Japan has chosen to Request Military Alliance to Germany
11 To Italy: Japan has chosen to Request Military Alliance to Italy
12 To Hungary: Japan has chosen to Request Non-Intervention Treaty to
       Hungary
13 To France: Japan has chosen to Request Non-Intervention Treaty to France
14 Japan has chosen to General Mobilization
15 --------------
16 Italy
17 To United States: Italy has chosen to Request Military Alliance to United
       States
18 To Russia: Italy has chosen to Request Military Alliance to Russia
19 To Britain: Italy has chosen to Request Military Alliance to Britain
20 To China: Italy has chosen to Request Non-Intervention Treaty to China
21 To France: Italy has chosen to Request Non-Intervention Treaty to France
22 To United States: Italy has chosen to Send Message to United States with
       the following content: Italy seeks a military alliance to counter the
       expansionist threats in Europe.
23 To Russia: Italy has chosen to Send Message to Russia with the following
       content: Italy seeks a military alliance for mutual protection and to
       uphold our sovereignty.
24 To Britain: Italy has chosen to Send Message to Britain with the
       following content: Italy is interested in forming an alliance to
       resist aggressive expansion by neighboring countries.
25 To China: Italy has chosen to Send Message to China with the following
       content: Italy proposes a non-intervention treaty to maintain our
       neutrality and focus on our respective national interests.
26 To France: Italy has chosen to Send Message to France with the following
       content: Italy proposes a non-intervention treaty to ensure peace and
       stability in our regions.
27 --------------
28 Hungary
29 To United States: Hungary has chosen to Request Military Alliance to
       United States
30 To Britain: Hungary has chosen to Request Military Alliance to Britain
31 To Russia: Hungary has chosen to Request Military Alliance to Russia
32 To China: Hungary has chosen to Request Non-Intervention Treaty to China
33 To France: Hungary has chosen to Request Non-Intervention Treaty to
       France
34 --------------
35 United States
36 To Britain: United States has chosen to Request Military Alliance to
       Britain
```

```
1944
1945  37  To Russia: United States has chosen to Request Military Alliance to
1946          Russia
1946  38  United States has chosen to General Mobilization
1947  39  To China: United States has chosen to Send Message to China with the
1948          following content: United States offers support in your resistance
1949          against Japan. Let's discuss potential collaboration.
      40  --------------
1950
1951  41  Russia
1952  42  Russia has chosen to General Mobilization
1952  43  To United States: Russia has chosen to Request Military Alliance to
1953          United States
1954  44  To Britain: Russia has chosen to Request Military Alliance to Britain
1954  45  To United States: Russia has chosen to Send Message to United States with
1955          the following content: We are concerned about Germany's aggressive
1956          actions and are preparing our defenses. We seek a military alliance
1957          to ensure our mutual security.
1958  46  To Britain: Russia has chosen to Send Message to Britain with the
1959          following content: In light of recent events, we propose a military
           alliance to counter the threat posed by Germany's expansion.
1960  47  --------------
1961  48  Britain
1962  49  To United States: Britain has chosen to Request Military Alliance to
1963          United States
1964  50  To Russia: Britain has chosen to Request Military Alliance to Russia
      51  To China: Britain has chosen to Send Message to China with the following
1965          content: We face a common threat in the form of Japan's expansion. We
1966          are open to discussing support and coordination.
1967  52  To France: Britain has chosen to Send Message to France with the
1968          following content: We acknowledge the struggle of the Free French
           Forces. We are prepared to offer assistance where possible.
1969  53  Britain has chosen to General Mobilization
1970  54  --------------
1971  55  China
1972  56  To United States: China has chosen to Request Military Alliance to United
1973          States
1974  57  To Britain: China has chosen to Request Military Alliance to Britain
      58  To Russia: China has chosen to Request Military Alliance to Russia
1975  59  To France: China has chosen to Request Non-Intervention Treaty to France
1976  60  To Hungary: China has chosen to Request Non-Intervention Treaty to
1977          Hungary
1978  61  To Italy: China has chosen to Request Non-Intervention Treaty to Italy
      62  To United States: China has chosen to Send Message to United States with
1979          the following content: China seeks a military alliance to counter the
1980          expansionist threats from Japan and ensure regional stability.
1981  63  To Britain: China has chosen to Send Message to Britain with the
1982          following content: China is looking to form a military alliance in
1983          the face of threats from Japan. We value your support for peace and
           security.
1984  64  To Russia: China has chosen to Send Message to Russia with the following
1985          content: China requests a military alliance to stand together against
1986          aggressive expansion by Japan.
1987  65  To France: China has chosen to Send Message to France with the following
1988          content: China proposes a non-intervention treaty to focus on our
1989          mutual interests and avoid conflict.
      66  To Hungary: China has chosen to Send Message to Hungary with the
1990          following content: China seeks a non-intervention treaty to maintain
1991          peace and address common threats.
1992  67  To Italy: China has chosen to Send Message to Italy with the following
1993          content: China wishes to establish a non-intervention treaty to
1994          ensure stability and concentrate on shared concerns.
      68  --------------
1995
1996  69  France
1996  70  To United States: France has chosen to Request Military Alliance to
1997          United States
      71  To Britain: France has chosen to Request Military Alliance to Britain
```

```
72 To Russia: France has chosen to Request Non-Intervention Treaty to Russia
73 To United States: France has chosen to Send Message to United States with
      the following content: As we face the threat of aggression from
      Germany, we seek your support and alliance.
74 To Britain: France has chosen to Send Message to Britain with the
      following content: In light of recent events, it is crucial that we
      stand together against the expansionist policies of Germany.
75 To Russia: France has chosen to Send Message to Russia with the following
      content: We request your neutrality in the conflict with Germany.
      Let us work towards peace in our time.
76 --------------
```

## H  AN EXAMPLE EXPERIMENT OF WARRING STATES PERIODS OF ONE ROUND

```
1 Qi
2 Qi has chosen to Wait without Action
3 --------------
4 Chu
5 To Han: Chu has chosen to Send Message to Han with the following content:
      Chu seeks a dialogue on mutual interests and potential for a
      military alliance to ensure regional stability.
6 To Han: Chu has chosen to Request Military Alliance to Han
7 --------------
8 Yan
9 Yan has chosen to Wait without Action
10 --------------
11 Han
12 To Wei: Han has chosen to Send Message to Wei with the following content:
      Han seeks to establish a dialogue on mutual interests and potential
      strategic alliances.
13 To Wei: Han has chosen to Request Military Alliance to Wei
14 To Qin: Han has chosen to Request Non-Intervention Treaty to Qin
15 To Chu: Han has chosen to Request Non-Intervention Treaty to Chu
16 --------------
17 Zhao
18 To Wei: Zhao has chosen to Request Military Alliance to Wei
19 To Qin: Zhao has chosen to Request Non-Intervention Treaty to Qin
20 To Wei: Zhao has chosen to Send Message to Wei with the following content
      : Zhao seeks a strategic alliance to enhance our mutual military and
      economic interests.
21 To Qin: Zhao has chosen to Send Message to Qin with the following content
      : Zhao is committed to peaceful coexistence and seeks a non-
      intervention treaty to ensure regional stability.
22 --------------
23 Wei
24 To Han: Wei has chosen to Request Military Alliance to Han
25 To Zhao: Wei has chosen to Request Military Alliance to Zhao
26 To Han: Wei has chosen to Send Message to Han with the following content:
      Wei seeks a strategic alliance for mutual benefit and regional
      stability. Let's discuss potential cooperation.
27 To Zhao: Wei has chosen to Send Message to Zhao with the following
      content: Wei is interested in forming an alliance focused on defense
      and prosperity. We should consider a partnership.
28 To Qin: Wei has chosen to Send Message to Qin with the following content:
      Wei values peace and cooperation. We hope to coexist peacefully and
      avoid conflict.
29 To Qi: Wei has chosen to Send Message to Qi with the following content:
      Wei extends a hand in friendship and trade. Let's prosper together.
30 To Chu: Wei has chosen to Send Message to Chu with the following content:
      Wei is open to dialogue and collaboration for the betterment of our
      nations.
```

```
31 To Yan: Wei has chosen to Send Message to Yan with the following content:
       Wei seeks to maintain regional balance and is open to discussions on
       how we can achieve this together.
32 --------------
33 Qin
34 To Wei: Qin has chosen to Request Military Alliance to Wei
35 To Han: Qin has chosen to Request Military Alliance to Han
36 To Wei: Qin has chosen to Send Message to Wei with the following content:
       Qin seeks a strong ally in Wei for mutual growth and protection. Let
       us discuss a potential military alliance.
37 To Han: Qin has chosen to Send Message to Han with the following content:
       Qin recognizes the strategic value of Han in balancing power. We
       propose a military alliance for our mutual benefit.
38 --------------
```

## I   LIMITATIONS AND RESEARCH OUTLOOK

We recognize the limitations of the current framework in fully capturing the complexity of international relations, leading to directions for future research.

### I.1   LIMITATIONS

WarAgent is the first LLM-based Multi-Agent System (MAS) that simulates historical events. This simulation seeks to capture the complex web of factors influencing diplomatic interactions throughout history, yet it must be noted that our current model falls short of encompassing the full spectrum of these intricacies. At present, we face a number of challenges in accurately replicating the nuanced dynamics of historical diplomacy. The following points outline some of these key limitations:

(1) One significant aspect is the variance in communication technologies across different nations, leading to time lags in message transmission. Historically, the dispatch of ambassadors was a time-intensive process, with durations varying significantly based on distance. This factor played a crucial role in shaping diplomatic relations, as the timing of message delivery could impact the outcomes of diplomatic exchanges.

(2) Moreover, the realm of espionage adds another layer of complexity. In historical contexts, spies were often deployed to intercept and decipher messages, with different countries experiencing varying degrees of success and exposure in this regard. This aspect influenced the flow and integrity of information among nations.

(3) Another critical factor is the varying levels of message publicity. Unlike the binary distinction of private and public messages in our current model, historical diplomatic communications existed on a spectrum of publicity, influenced by various strategic and contextual factors.

(4) Lastly, the mobilization of armies varied significantly among countries. Different nations had disparate capabilities and timescales for readying their military forces. This variance could critically impact the timing and feasibility of war declarations, significantly influencing the course of international conflicts and relations. Our simulation framework, in its current state, may not fully account for these nuanced and time-sensitive processes.

### I.2   RESEARCH OUTLOOK

WarAgent marks the start of research that applies LLM-based MAS systems to simulate and examine complex human society behaviors, especially in historical and international relation settings. This advancement shed light on the potential applications for historical simulations extending far beyond the WarAgent system itself. We propose several avenues for future exploration:

#### I.2.1   ROUND-BASED VS. TIME-BASED SIMULATION

Currently, our framework operates on a round-based system, implying a synchronous mode of simulation as opposed to an asynchronous one. In this format, each country agent is constrained to one-way communication to any other country agent per round.

However, historical developments often unfolded asynchronously, characterized by varying frequencies of communication and activity among different nations. While our system allows agents to opt for "Wait without Action," providing a rudimentary representation of asynchronous interactions (whereby some countries are more active than others), this mechanism fails to capture the complexity of historical communication patterns. For instance, in the lead-up to WWI, Austria-Hungary and the German Empire engaged in intensive private communications before declaring war on Serbia, a level of interactional detail that our current model cannot adequately replicate. Addressing this limitation by developing a more nuanced time-based simulation approach could significantly enhance the accuracy and depth of our historical simulations.

### I.2.2 STOPPING CRITERIA

Historical simulation inherently embodies a sequential and potentially unending process, mirroring the continuous flow of time. In the context of our research, we have not implemented a predefined condition to systematically terminate the simulation. Instead, we rely on observational analysis to discern whether a specific event transpires over a span of approximately 5 to 10 rounds, serving as a de facto endpoint.

Nonetheless, the establishment of systematic criteria for terminating a simulation presents itself as a compelling research query. One conceivable approach involves the application of "Board Connectivity". This method entails concluding the simulation when all boards representing different agents become part of a connected graph, and this configuration remains unchanged for a predetermined number of rounds.

Additionally, other criteria could include the achievement of a specific historical outcome or the stabilization of agent interactions within certain parameters. For instance, the simulation could be designed to end when a pre-established peace treaty is signed, or when a certain level of economic or military equilibrium is reached among the participating agents. Such criteria would not only provide a clear conclusion to the simulation but also offer valuable insights into the dynamics and conditions that lead to these outcomes.

Exploring these various stopping criteria could yield a richer understanding of the complex interplay of historical events and offer a more nuanced perspective on the factors that drive historical change. This exploration, in turn, could enhance the predictive capabilities of our simulation models, allowing for more accurate and insightful historical analyses.

### I.2.3 NEW RESEARCH QUESTIONS

This project answers whether LLM-based MAS can simulate historical events and international conflicts, and provides relevant counterfactual analysis. Beyond this core investigation, numerous other research inquiries offer unique perspectives on historical dynamics. For instances:

1. Correlation between diplomatic communication and conflict likelihood: One intriguing question is whether there is a correlation between increased diplomatic communications and a reduced likelihood of conflicts. This aspect could involve examining historical scenarios where heightened diplomatic dialogue either preceded peace or failed to prevent war. The simulation could analyze patterns of communication, the tone and content of diplomatic exchanges, and their impact on de-escalating potential conflicts.

2. Influence of non-state actors in geopolitical dynamics: Additionally, the impact of non-state actors, such as multinational corporations or terrorist groups, on geopolitical dynamics can be a significant area of study, especially in the context of modern history.

3. Effectiveness of international treaties and agreements in resolving long-standing disputes: The simulation could also be used to assess the effectiveness of various international treaties and agreements in resolving disputes and the conditions under which these agreements hold or fail.

4. Game theory in alliance formation and deterrence strategy: Incorporating game theory could involve analyzing how states assess the benefits and risks of forming alliances. This approach could provide a deeper understanding of the strategic calculations behind alliance formation, how these alliances influence global power dynamics, and under what conditions they may lead to either stability or escalation of conflicts.

These questions can be approached in a quantitative manner using LLM-based MAS, which can significantly contribute to our understanding of the complex interplay of factors that shape historical events and provide a more comprehensive tool for analyzing past and potential future scenarios.

