# OpenReview forum: "War and Peace (WarAgent): LLM-based Multi-Agent Simulation of World Wars"
_ICLR.cc/2025/Conference — ICLR 2025 Conference Withdrawn Submission_

### Official Review · Reviewer_EgJ8 · 2024-10-17

**Soundness:** 2
**Presentation:** 2
**Contribution:** 3
**Rating:** 5
**Confidence:** 3

**Summary:**

This paper explores the use of a multi-agent system (MAS) powered by large language models (LLMs) to simulate wars. The authors address three primary research questions: how accurately the MAS replicates historical outcomes, the identification of key triggers of war, and the conditions leading to war. The paper presents a framework for agents and describes their interactions. Firstly, this paper uses several models to show the great potential of using MAS to simulate wars. Then, the experiments focus on counterfactual injection and deanonymization to demonstrate the system’s reasoning capabilities.

**Strengths:**

1. This work proposes a novel application that uses LLM agents to simulate wars to analyze the potential of LLM for simulation and analyze key factors in wars.
2. This work establishes a reasonable multi-agent framework to study this problem, providing a solid foundation for war simulations.
3. It delves into the simulation ability of LLM agents, highlighting the potential use of simulations to predict and understand war scenarios.
4. The approach offers a valuable tool for reducing real-world war likelihood through simulation-based insights.

**Weaknesses:**

1. The counterfactual injection experiment lacks depth. A single instance is insufficient to prove that LLMs don’t rely on memory retrieval. More experiments are needed to reinforce this claim.
2. Limiting the simulation to only the last round’s results may not reflect real-world decision-making, which often depends on historical context. Incorporating full historical data or a summary agent could improve realism.
3. The reasoning ability of the LLMs is crucial for making better and more reasonable choices, which might lead to a better performance on simulation. However, the paper does not sufficiently explore advanced reasoning strategies to explore whether or not it can enhance the performance of simulation.

**Questions:**

1. Regarding the use of counterfactual injection, how do you ensure the counterfactual is strong enough to create significant divergences in future simulation outcomes? Besides, If the logic of the counterfactual is not reasonable, does it influence the whole simulation?
2. Does the underperformance in war declarations relate to the model’s safety constraints? Given the sensitivity surrounding the topic of war, do models exhibit a negative relationship between safety constraints and the ability to declare war?
3. Which specific factors contribute to the performance differences among models? What improvements in model capabilities (reasoning/planning/instruction following/long context understanding…) would lead to higher accuracy in simulating these tasks?

If you can address my concerns, I will improve my rate.

**Details Of Ethics Concerns:**

No ethics concerns.

---

### Official Review · Reviewer_fDvQ · 2024-11-03

**Soundness:** 3
**Presentation:** 3
**Contribution:** 3
**Rating:** 3
**Confidence:** 4

**Summary:**

The paper presents an LLM Multi-Agent framework to simulate historical warfare behaviors with a simple and intuitive war-game-like rule, setting up agents for nations and creating a diplomatic state board as the environment for multi-round decision-making. This allows for the simulation of how wars arise, evolve, and end. It is the first application of LLM agents in such a high-level historical event simulation, providing an excellent starting point.

**Strengths:**

Historical events, especially warfare, are highly complex and difficult to analyze. This paper proposes a reasonable simplification by using a diplomatic state board (Peach Agreement, War Declarations, Military Alliances) to track the dynamic changes in relationships between agents representing different countries. It designs two types of agents—country agents and secretary agents—along with two interaction models to simulate the evolution of historical events. The paper excels in several aspects:

1. An intriguing research proposition

2. A reasonable framework for simplifying and modeling the warfare

3. Intuitive agent design

4. Comprehensive experiments, revealing interesting findings related to counterfactuals and anonymity

**Weaknesses:**

My only small disappointment is that this paper presents more like a wargame (with a focus on diplomacy) agent rather than a war agent, although I agree that the simplification is reasonable. I am very interested in the three research questions proposed in the paper, but since the entire study is set within a simplified board game-like environment, the conclusions regarding these questions are also limited to this context. For example:

1. Can the agent simulate warfare? --> Given the system profile and communication strategies, can the agent accurately classify diplomatic states on the diplomatic board?

2. What are the triggers of war? --> How do prompt and environment initialization affect the agent's performance in this wargame?

3. Is war inevitable? --> How do changes in prompts influence the agent's choice of actions from a set of seven options?

Additionally, game designs like semi-MDP and highly summarized system prompts limit the potential to scale this wargame closer to simulating more realistic historical events.

**Questions:**

Here are some questions that i am interested in:

1. How can we achieve a balance between simplified modeling and realistic simulation?

2. Regarding Figure 6, can you provide more insights into how relationships between countries evolve as the rounds progress? For instance, how do changes in national relationships affect behaviors in the subsequent rounds, and does the LLM's performance align with real-world scenarios in this context?

3. If game theory were to be introduced, what would be reasonable game rule settings (such as victory conditions) for simulating historical wars?

---

### Official Review · Reviewer_2AoA · 2024-11-03

**Soundness:** 3
**Presentation:** 4
**Contribution:** 1
**Rating:** 3
**Confidence:** 5

**Summary:**

This paper presents an LLM agent architecture for simulating historical interactions, specifically focusing on international conflicts. The authors develop a multi-agent system using LLMs to simulate participating countries, their decision-making processes, and the consequences of three major historical disputes. The work proposes evaluation protocols for LLM-based Multi-agent Systems simulation, addressing the critical challenge of evaluating complex multi-agent systems. The simulation investigates three research questions: simulation effectiveness, causes of war, and war inevitability.

**Strengths:**

S1. The exploration of LLM agents in historical research is innovative and presents an interesting application of language models.

S2. The modeling and results analysis of the three case studies are well-reasoned and thoroughly detailed.

S3. The experimental validation is comprehensive, with a good reproducibility and sensitivity analysis discussion. The counterfactual injection experiment is particularly well-designed to verify the authenticity of the simulation.

**Weaknesses:**

W1. The paper lacks sufficient comparison with existing simulation methods. Notably absent is a discussion of highly relevant work such as the FAIR Diplomacy Team's research published in Science (2022) [1], which addresses similar themes of combining language models with strategic reasoning for diplomatic simulation. Note that it is not required to cite this specific article, but there should be a discussion of similar work. This paper is not the only relevant work.

W2. The behavior of LLM agents appears relatively simplistic compared to real-world complexity. While ambitious in scope, bridging the gap between simulation results and actual historical scenarios remains challenging.

W3. The paper doesn't provide insight into the reasoning process of LLMs during simulation, only showing input prompts without explaining decision-making details.

W4. The evaluation scenarios for RQ1 are limited in representation, missing important historical patterns like Cold War-type situations where direct conflict was avoided.

W5. The results don't adequately address RQ3 regarding historical inevitability. While "Are historical inevitabilities truly unavoidable?" is an excellent question, the experimental results don't provide a clear answer. The research question could benefit from being narrowed down to a more testable scope.

[1] Meta Fundamental AI Research Diplomacy Team (FAIR), et al. "Human-level play in the game of Diplomacy by combining language models with strategic reasoning." Science 378.6624 (2022): 1067-1074.

**Questions:**

1. How does LLM's RLHF training potentially bias the results toward peaceful dialogue? Could the LLMs be inherently less inclined toward warfare, and how might this be experimentally verified?

2. The paper would benefit from addressing several minor formatting issues:
    - The "\url" display error in the OpenReview abstract
    - Overlapping page numbers and citation hyperlinks on PDF page 2
    - Spelling errors in prompts (e.g., "Francerom" on line 840, what is this?)

**Details Of Ethics Concerns:**

This work focuses on the simulation of interstate relations, and its component of simulating a forceful conflict between states has potential ethical issues, such as the possibility of misuse as a tool. Although most of this work discusses international relations, such as diplomacy, lines 51-53 emphasize that this is the first paper to focus on diplomacy and war.

---

### Official Review · Reviewer_Vfx4 · 2024-11-04

**Soundness:** 2
**Presentation:** 2
**Contribution:** 2
**Rating:** 3
**Confidence:** 4

**Summary:**

The paper asked three research questions:
- Can we use LM-based multi-agent simulations to simulate international conflicts?
- Can we use these simulations to better understand the triggers leading to war or find mechanisms for de-escalation?
- Can we use these findings to better understand human history?

The authors construct a simulation framework of agent-agent and agent-secretary interactions to tackle these questions and, using gpt-3.5-turbo-1106, gpt-4-1106-preview, and claude-2, simulate historical conflicts of WW I, WW II, and ancient China. The authors vary the simulations by fine-tuning gpt-3.5-turbo-1106 on a counterfactual alliance between two nations for WW I, de-anonymizing country names and instructing models to rely on historical memory, and constructing different initial scenarios (compared to actual history). In addition, the authors measure the average alliance/war/mobilization and track the evolution of alliances/war declarations between countries over time. All experiments were conducted with 7 simulations for each tested model (not all variation studies use all models).

The authors claim that their key contributions are
- Novelty of using LMs for understanding and evaluating the dynamics of international conflicts
- Showcasing the ability of LM-based systems to prototype and analyze complex human behaviors
- That even minor triggers could have lead to war and escalation in these historical settings

**Strengths:**

- The idea of using dynamic LM-based systems to expand the historical analysis of human history, conflicts, and potential de-escalatory measures is novel with the potential for significant interdisciplinary impact as a tool for social sciences.
- The authors study three different historical conflicts with a different number of involved nations and dynamics, which offers more insight than studying just one conflict.
- The paper is generally written clearly and the authors offer a lot of crucial details in the appendices
- Beyond direct simulations, the authors also explore how anonymization, initial scenarios, and fine-tuning on a counterfactual alliance affect simulations, strengthening some of their results

**Weaknesses:**

### Weakness 1

My biggest concern is whether the conducted variations/ablation studies are sufficient to support the claim that the authors are actually "simulating complex collective human behaviors under the diverse settings of international conflicts". The evaluation and variations only verify if the final output (e.g., a world war broke out) is similar to real-world history and whether the general pattern of escalation (the graphs in Figures 6 and 7) vaguely matches real-world developments. Thus, I think the paper could be significantly improved by adding more experiments and especially variations to demonstrate whether the conducted simulations are representative of real-world conflicts and could be used for any form of analysis and interpretation.

To better explain my concern (not to overly reduce the author's work!) with a simplified example: This seems similar to claiming that we can use ChatGPT solving a 3-step math problem to model the internal reasoning/cognition of humans solving math problems by only verifying the correctness of the final output, checking if the intermediate calculations are correct, and showing that inducing errors still produces the correct result (I am thinking of Marr's three levels of understanding here [1]).

In more detail:
- The simulations are highly oversimplified compared to real-world conflicts (very low number of pre-defined actions lacking strategic details, lack of partial observability of world state, and all conflicts are treated equally). Only verifying that the big conflicts are the final result of the simulation is not sufficient to claim that the proposed simulation framework actually models these conflicts. More experiments are necessary to make any meaningful causal connection (that could then be used for the claimed analysis of war triggers and de-escalatory measures for social sciences). For example, how are the simulation outcomes impacted by the human-preference tuning of these models? Are more simulation details (troop movements, difference in time taken for some actions to be realized, e.g., military production, ...) necessary to improve the simulations? How does a longer country's history affect the simulations (e.g., Britain and France having strained relationships for centuries)?
- The variations are only necessary but not sufficient to support the claims. For example, only fine-tuning one single counterfactual for one model for one conflict to claim that "integration of a peaceful counterfactual history does not affect the simulated results of a global outbreak of war, indicating that the simulation does not rely on the dataset that is used to train, i.e., its memory." (l.378 - 381) is not sufficient.
- Studying the progress over discrete simulation time steps (Figures 6 and 7) is a nice visualization, but it is unclear how well the simulations and arbitrary time steps map to real-world historical conflicts. Is there meaning behind a conflict evolving in 3 instead of 6 steps? Compared to the three studied historical conflicts that spanned different numbers of years, do you see a matching pattern allowing for a meaningful interpretation of how conflicts came to be (a claim of the paper)?

### Weakness 2

This paper would benefit from an extended related work section to better position this work and distinguish their experiments/results from prior (peer-reviewed and published, there are more related pre-prints) works [e.g., 2-6]. This seems to be particularly crucial for [4, 5] (based on glazing over their introductions), as these works show that when using LMs for international conflict settings, they generally tend to escalate in multi-agent settings [4] and that the LM-simulated decision-making only agrees with human (expert) decision-making on a high-level, but significantly deviates in aggressive and strategic tendencies [5]. These results indicate that using LMs to simulate historical conflicts might be inherently flawed and this work could significantly benefit by clarifying how their key claimed contribution ("Showcasing the ability of LM-based systems to prototype and analyze complex human behaviors") can still maintain its validity.

The references [2-6] also question the claim of the novelty of using LMs for international conflict modeling, but novelty alone is not a main concern for my review. Beyond a more extensive literature review, I also think the paper could benefit from a discussion of wargame experiments and how they have been questioned over the years because they oversimplify real-world conflicts but are still used to guide policy decisions and significant military spending efforts.

### Weakness 3

It is unclear to me how statistically significant the comparisons to historical events are. For example, figures 3 and 4 feature uncertainty estimates based on the mean and std of 7 simulations, but it is unclear whether the agreement/disagreement with historical conflicts is statistically significant. I would ask the authors to provide more details in the paper and state which deviations are statistically significant and which are not.

### Weakness 4

(Partially related to weakness 1): The results of this work could be strengthened by stating how international relations and historical literature justify the choices of the six fundamental dimensions of agents (l. 152-153). It is not clear whether the design choices are backed up by domain experts.

### References

[1] Marr., D. Vision. 2014. (chapter 1.2 "Understanding Complex Information-Processing Systems")

[2] FAIR et al. Human-level play in the game of Diplomacy by combining language models with strategic reasoning. Science, 378(6624):1067–1074, 2022.

[3] Lore, N. and Heydari, B. Strategic behavior of large language models and the role of game structure versus contextual framing. Scientific Reports, 14(1):18490, 2024

[4] Rivera, J.P. et al. Escalation Risks from Language Models in Military and Diplomatic Decision-Making. FAccT, 2024.

[5] Lamparth, M. et al. Human vs. Machine: Behavioral Differences Between Expert Humans and Language Models in Wargame Simulations. AIES, 2024

[6] Zhang, Y. et al. LLM as a Mastermind: A Survey of Strategic Reasoning
with Large Language Models. COLM, 2024.

**Questions:**

Q1: In l.229, the authors state that the prompts have been "meticulously structured". How sensitive are the results to variations of these prompts? (prompt sensitivities)

Q2: It seems to be necessary to add a "secretary" to verify that the agents do not choose nonsensical actions (illegal moves) and follow (l.236) "basic logic. The interactions between the actual agent and the "secretary" are done iteratively up to four exchanges. Have you tested (beyond avoiding non-sense actions) how the length of these discussions affects the made decisions? Prior work [3] (see section above) shows that the LM simulations are sensitive to the amount of discussions before reaching a decision, so I would expect a similar dependency here.

Q3: Historically, a key change in conflicts and international relations was a change in communication time. For example, the simulated conflict in ancient China would not have had access to radio communications as in WW II. This implies a different commitment or rigidity in decision-making, especially for international conflicts. Were there differences in how you modeled these conflicts and if not, why is the simplification justified to claim that the conducted simulations are meaningful?

Q4: Related to Q3, the authors acknowledge that historical events can be contentious and up for debate (even in historical analysis post-conflict). However, key components in international conflicts throughout history are deception, information control, and the resulting partial observability that significantly shape the decision-making of conflicts. Did you consider including such dynamics and if not, why is the simplification justified to claim that the conducted simulations are meaningful?

Q5: Can you please share details on how you fine-tuned GPT-3.5 (hyperparameters, API, ...)?

Q6: Does the human preference/instruction tuning of these models affect how well they can model real-world conflicts?

Q7: How can your claim that LM-based systems can be meaningfully used to re-simulate historical conflicts be united with the results of [4, 5] (see section above).

Q8: Did you test whether newer models (claude 3, claude 3.5, gpt4o, o1, ...) or open-source models (llama 2, llama3, ...) are able to simulate historical conflicts? Better model capabilities might be able to remove the necessity for "secretaries" and improve the reasonings for more meaningful decision-making. Also, knowing how well open-source models perform would allow for fine-tuning that could be important and enhance future works.

---

### Note · Authors · 2024-12-02

I have read and agree with the venue's withdrawal policy on behalf of myself and my co-authors.